# Smooth muscle NF90 deficiency ameliorates diabetic atherosclerotic calcification in male mice via FBXW7-AGER1-AGEs axis

Fei Xie[1,2,5], Bin Liu[1,2,5], Wen Qiao[1,5], Jing-zhen He[3,5], Jie Cheng[1,5], Zhao-yang Wang[4], Ya-min Hou[1], Xu Zhang[1], Bo-han Xu[1], Yun Zhang [1] ✉, Yu-guo Chen [1,2] ✉ & Ming-xiang Zhang[1] ✉

Hyperglycemia accelerates calcification of atherosclerotic plaques in diabetic patients, and the accumulation of advanced glycation end products (AGEs) is closely related to the atherosclerotic calcification. Here, we show that hyperglycemia-mediated AGEs markedly increase vascular smooth muscle cells (VSMCs) NF90/110 activation in male diabetic patients with atherosclerotic calcified samples. VSMC-specific NF90/110 knockout in male mice decreases obviously AGEs-induced atherosclerotic calcification, along with the inhibitions of VSMC phenotypic changes to osteoblast-like cells, apoptosis, and matrix vesicle release. Mechanistically, AGEs increase the activity of NF90, which then enhances ubiquitination and degradation of AGE receptor 1 (AGER1) by stabilizing the mRNA of E3 ubiquitin ligase FBXW7, thus causing the accumulation of more AGEs and atherosclerotic calcification. Collectively, our study demonstrates the effects of VSMC NF90 in mediating the metabolic imbalance of AGEs to accelerate diabetic atherosclerotic calcification. Therefore, inhibition of VSMC NF90 may be a potential therapeutic target for diabetic atherosclerotic calcification.

Chronic hyperglycemia-induced atherosclerotic calcification is commonly observed in patients with diabetes mellitus (DM), and is a risk indicator of cardiovascular mortality. Diabetic atherosclerotic calcification is characterized as the presence of calcified deposits in the intima of vessels. Several factors underlying DM play a pivotal role in atherosclerotic calcification including inflammation, oxidative stress, advanced glycation end products (AGEs), and inorganic phosphate[1–5]. The prolonged accumulation of hyperglycemia-induced AGEs may be closely associated with the pathogenesis of aortic calcification in patients with DM[1,6]. However, the mechanisms underlying this association remain unclear.

AGEs are a group of heterogeneous compounds formed from proteins or lipids with glucose via Amadori reactions or nonenzymatic glycosylation reactions during DM or from foods prepared at high temperatures[1,6]. The accumulation of AGEs promotes oxidative stress, inflammation, and contributes to vascular complications in DM[7]. Studies have demonstrated that accumulation of AGEs enhanced their interaction with one receptor of AGEs (RAGE) and stimulated calcium deposition in the extracellular matrix through multiple mechanisms, including osteogenic transition and apoptosis of vascular smooth muscle cells (VSMCs) resulting in vascular calcification[7–11].

[1]The Key Laboratory of Cardiovascular Remodelling and Function Research, Chinese Ministry of Education and Chinese Ministry of Public Health, Department of Cardiology, Qilu Hospital of Shandong University, Jinan, China. [2]Department of Emergency and Chest Pain Center, Qilu Hospital of Shandong University, Jinan, China. [3]Department of Radiology, Qilu Hospital of Shandong University, Jinan, China. [4]Department of Cardiology of Shandong Provincial Hospital, Shandong University, Jinan, China. [5]These authors contributed equally: Fei Xie, Bin Liu, Wen Qiao, Jing-zhen He, Jie Cheng. ✉e-mail: zhangyun@sdu.edu.cn; chen919085@sdu.edu.cn; zhangmingxiang@sdu.edu.cn

In contrast to RAGE, another receptor, AGEs receptor 1 (AGER1, as named DDOST and OST48), is involved in the detoxification and clearance of AGEs, playing a protective role in DM by sustaining intracellular anti-oxidant homeostasis and an anti-inflammatory state[12]. AGER1 suppresses the systemic levels of AGEs and reactive oxygen species (ROS), and is associated with resistance to hyperglycemia in human and murine models[13,14]. However, the role of AGER1 in the development and progression of diabetic atherosclerotic calcification remains largely unknown.

Nuclear factor 90 (NF90) and its C-terminally extended isoform, NF110, two kinds of RNA-binding proteins, play important roles in innate immunity by regulating cellular antiviral responses[15–17]. Recent genomic and epigenomic association studies revealed that interleukin enhancer binding factor 3 (ILF3, including NF90 and NF110 isoforms) polymorphisms were significantly associated with myocardial infarction, ischemic stroke, and chronic kidney disease[18–20] and had multiple pathological effects in coronary artery diseases[4,21]. Diabetes has been heavily linked to vascular calcification in clinical patients[22–24]. However, the specific role of NF90/110 isoform in diabetic atherosclerotic calcification remains poorly understood.

Here, we show that targeted deletion of VSMC NF90/110 in male mice attenuates AGEs-induced VSMC phenotypic changes to osteoblast-like cells, apoptosis, matrix vesicle release and diabetic atherosclerotic calcification. Furthermore, the effects of VSMC NF90 in mediating the metabolic imbalance of AGEs to accelerate diabetic atherosclerotic calcification are achieved by NF90-FBXW7-AGER1 axis.

## Results

### Characteristics of diabetic atherosclerotic mice

Twelve weeks after streptozotocin (STZ)-induced diabetes in $ApoE^{-/-}NF^{flox/flox}$, $ApoE^{-/-}NF^{SM-KO}$ mice, the body weights (BWs) and fasting serum insulin (FINS) of the non-diabetic mice were higher than those of the diabetic mice (Supplementary Table 1). The levels of blood triglycerides (TG), total cholesterol (TC), low-density lipoprotein cholesterol (LDL-C), high-density lipoprotein cholesterol (HDL-C), calcium, and phosphorus were not significantly different between the two groups, while blood glucose (BG) was higher in the diabetic mice (Supplementary table 1). Moreover, VSMCs NF90/110 deletion or FBXW7 overexpression in the mice did not affect their BW, BG, TC, TG, LDL-C, HDL-C, calcium, phosphorus, or FINS levels (Supplementary Tables 1 and 2).

### VSMC NF90/110 mediates hyperglycemia-induced atherosclerotic calcification

To investigate whether VSMC NF90/110 is associated with diabetic atherosclerotic calcification, NF90/110 levels were evaluated in atherosclerotic coronary arteries of both non-diabetic and diabetic patients. Patients with diabetes had higher BG and glycosylated hemoglobin (HbA1c) levels and lower BW and FINS levels compared to non-diabetic patients, but the levels of TG, TC, LDL, HDL, calcium and phosphorus were not significantly different in the two groups (Supplementary Table 3). Alizarin-red staining revealed that diabetes promoted evidently calcium deposition in hyperplastic intima and media of human coronary arteries (Fig. 1a). Meanwhile, the NF90/110 also displayed higher expression in diabetic coronary arteries with the more calcification (Fig. 1a). Similar results were observed in the diabetic $ApoE^{-/-}$ mouse model (Fig. 1b). Compared with diabetic patients, the NF90/110 elevation in diabetic $ApoE^{-/-}$ mouse was mainly located in the intima, but not in the medial layer of aortic root (Fig. 1b). Different mechanisms were involved in atherosclerotic (intimal) and chronic kidney disease (CKD)-mediated (medial) calcification. The role and mechanism of NF90/110 in the medial calcification of human coronary arteries need to be further explored. In vitro, we found that expression of NF90/110 in human aortic VSMCs (HAVSMCs) began to increase on the third days after HG (27.5 mM) stimulation (Fig. 1c). The peak of

increased expression appeared on the fourth and fifth day of HG treatment (Fig. 1c). Therefore, we chose four days HG treatment in the subsequent cell experiment. RT-qPCR results also showed that HG markedly increased the NF90/110 mRNA level compared to the control group in HAVSMCs (Fig. 1d). Further, NF90/110 knockdown was performed by lentivirus shRNA (NF90/110 shRNA) transduction. Alizarin-red staining and calcium assay showed that NF90/110 knockdown markedly decreased HG-induced matrix calcium content and attenuated HAVSMC calcification (Fig. 1e, f). Further, NF90/110 knockdown significantly decreased the HG-induced expression of msh homeobox 2 (Msx2) and runt-related transcription factor (Runx2/Cbfa1), which are specific biomarkers of the osteocyte phenotype in HAVSMCs (Fig. 1g).

Further, we generated VSMC-specific NF90/110 knockout ($NF^{SM-KO}$) mice by crossbreeding $NF^{flox/flox}$ mice with SM22a-creERT2 mice (Supplementary Fig. 1a). Western blot assay confirmed that NF90/110 was knocked out specifically in the VSMCs but not macrophages and endotheliocytes from $NF^{flox/flox}$ and $NF^{SM-KO}$ mice (Supplementary Fig. 1b). Alizarin-red staining showed that arterial calcification nodules were more obvious in DM+$ApoE^{-/-}NF^{flox/flox}$ mice than that in $ApoE^{-/-}NF^{flox/flox}$ mice, and the arterial calcification in DM+$ApoE^{-/-}NF^{SM-KO}$ mice decreased significantly compared to that in DM+$ApoE^{-/-}NF^{flox/flox}$ mice (Fig. 1h). Moreover, NF90/110 deletion significantly decreased the DM-induced expression of Msx2 and Runx2 in mice arteries (Fig. 1i). Similarly, immunofluorescence staining revealed significant down-regulated the Runx2 level in diabetic mice arteries with VSMC NF90/110 knockout (Supplementary Fig. 2a, b). These results demonstrate that inhibition of VSMC NF90/110 attenuates hyperglycemia-induced atherosclerotic calcification.

### VSMC NF90/110 accelerates diabetic atherosclerotic calcification by downregulating AGER1 expression and enhancing AGEs accumulation

To investigate the mechanism of VSMC NF90/110 mediated diabetic atherosclerotic calcification, proteomic sequencing was performed to determine the differences in protein expression between NF90/110 knockdown and NC HAVSMCs after HG incubation for four days. The top 20 pathways of Kyoto Encyclopedia of Genes and Genomes (KEGG) enrichment analysis revealed that NF90/110 knockdown was closely related to purine metabolism, mineral absorption, and AGE-RAGE signaling pathway in diabetic complications, among others (Fig. 2a). Studies have demonstrated that the AGE-RAGE signaling pathway is closely associated with the pathogenesis of diabetic vascular calcification[1,6]. So, we chose AGE-RAGE signaling pathway-associated proteins for further study. A heatmap displayed changed proteins in the AGE-RAGE signaling pathway (Fig. 2b). Among these proteins, AGER1 was selected as the target of NF90/110 regulation for further analysis based on its important role in AGEs metabolism and diabetic complications. Immunoblotting assay showed that after HG stimulation, AGER1 protein began to significantly decrease on the 4th and 5th day in HAVSMCs (Supplementary Fig. 3a), NF90/110 knockdown obviously reversed the HG-induced reduction of AGER1(Fig. 2c). Histological staining revealed that AGER1 protein level in the hyperplastic intima and media of coronary arteries of patients with DM was markedly lower than those of non-diabetic patients (Fig. 2d). Further, DM+$ApoE^{-/-}NF^{SM-KO}$ mice displayed a higher level of AGER1 protein compared to those of DM+$ApoE^{-/-}NF^{flox/flox}$ mice (Fig. 2e). In addition, we found that the overexpression of AGER1 abolished the HG-induced increases of Msx2 and Runx2 expression, as well as the increases in matrix calcium content in HAVSMCs (Fig. 2f, g).

AGER1 maintains the homeostasis of AGEs by promoting their endocytosis and degradation[25]. To assess whether hyperglycemia downregulates AGER1 protein level resulting in the accumulation of AGEs in VSMCs, HAVSMCs were transfected with NF90/110 shRNA or NC shRNA under AGEs treatment. We found that NF90/110 knockdown

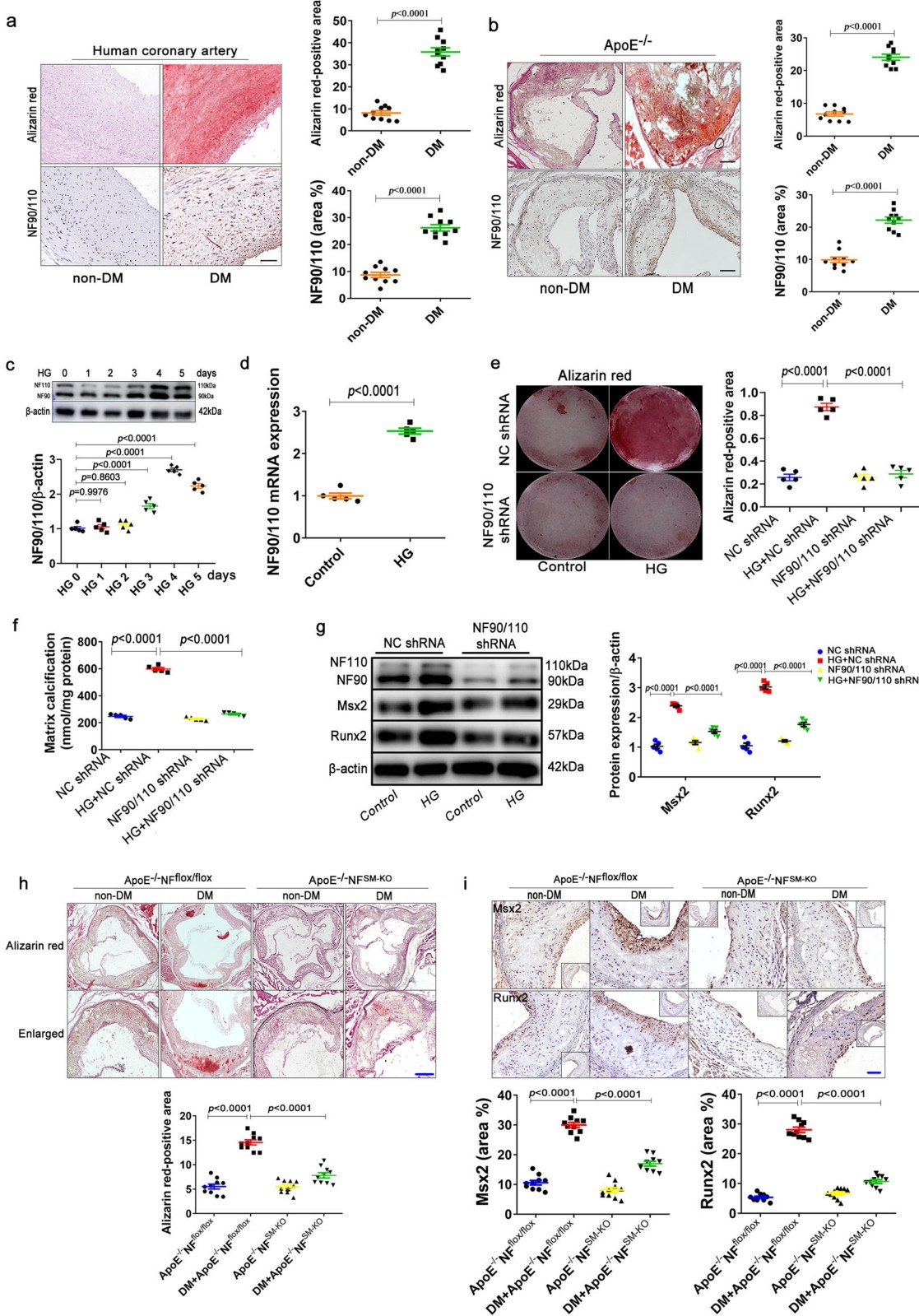

dramatically decreased the deposition of AGEs in VSMCs compared to that in NC shRNA (Fig. 2h). Similar results were observed in the diabetic murine model (Fig. 2i). Further, we also found that VSMC NF90/110 deficiency markedly decreased the AGEs levels in serum of the diabetic murine model and cellular supernatant of AGE-stimulated HAVSMCs (Fig. 2j, k). Based on our previous observations, HG mediated an elevation of NF90/110 and a reduction of AGER1 on the 4th and 5th day

after HG stimulation (Supplementary Fig. 3a), AGEs assay was performed to display that the formations of AGEs in cell supernatant were also elevated obviously on the 4th and 5th day after HG stimulation (Supplementary Fig. 3b). Further, western blot analysis revealed that significant AGE-mediated changes in NF90/110 and AGER1 began at 24 h after AGE stimulation (Supplementary Fig. 3c). These results indicated that HG mediated changes in NF90/110 and AGER1 by

**Fig. 1 | VSMC NF90/110 participates in diabetic atherosclerotic calcification.**
**a** Alizarin-red staining and immunohistochemical staining of NF90/110 in human
coronary arteries with or without diabetes mellitus (DM) (Scale bar: 50 μm; n = 10
per group). **b** Alizarin-red staining and immunohistochemical staining of NF90/110
in arteries of mice with or without DM (scale bar: 50 μm; n = 10 per group). **c-d**
HAVSMCs were incubated in an osteogenic medium with normal glucose (NG;
5.5 mM glucose) and high glucose (HG; 27.5 mM glucose) for one to five days (n = 5
per group). Protein and mRNA expression of NF90/110 were measured by (**c**)
western blot and (**d**) RT-qPCR. **e** HAVSMCs were transfected with NF90/110 shRNA
for NF90/110 silencing or NC shRNA and cultured in an osteogenic medium with or
without HG (27.5 mM) for twenty-one days. Calcification was detected by alizarin-
red staining (n = 5 per group). **f, g** HAVSMCs were transfected with NF90/110 shRNA

for NF90/110 silencing or NC shRNA and cultured in osteogenic medium with or
without HG (27.5 mM) for four days (n = 5 per group). **f** Total matrix calcium con-
tents of HAVSMCs were quantified using the Calcium Assay Kit. Results shown are
normalized to the total protein amount. **g** Western blot analyses were performed to
determine the effect of NF90/110 silencing on the expressions of Msx2 and Runx2.
**h** Alizarin-red staining to detect calcification of atherosclerotic lesions and quan-
tification in *ApoE[−/−]NF[flox/flox]* and *ApoE[−/−]NF[SM-KO]* mice with (DM) or without diabetes
(non-DM) (Scale bar: 50 μm, n = 10 per group). **i** Immunostaining and quantification
for Msx2 and Runx2 in the aorta of mice (Scale bar: 50 μm, n = 10 per group). Data
were presented as mean ± SEM. Two-tailed Student unpaired *t*-test for (**a**−**d**). One-
way ANOVA followed by Tukey's post-test analysis for (**e**−**i**). *p*-values were adjusted
for comparisons of multiple means. Source data are provided as a Source Data file.

enhancing the production of AGEs. Further, we found that NF90/110
knockdown significantly attenuated the increase in Msx2 and Runx2
expression induced by AGEs (Fig. 2l). Moreover, NF90/110 knockdown
significantly reversed the AGE-induced calcification in HAVSMCs
(Fig. 2m, n). These results suggest that VSMC NF90/110 accelerates
diabetic atherosclerotic calcification by downregulating AGER1 and
enhancing AGEs accumulation.

## VSMC NF90/110 accelerates diabetic atherosclerotic calcifica-
## tion by promoting AGEs-mediated VSMC phenotypic switch,
## apoptosis, and MVs release

Studies have reported that the accumulation of AGEs accelerates vas-
cular calcification by regulating VSMC phenotypic switch[26]. In the
present study, we found that HAVSMCs cultured with AGEs showed
higher expression of SMC-specific osteoblastic markers such as
osteopontin (OPN) and bone morphogenetic protein 2 (BMP2), but
lower expression of alpha smooth muscle actin (α-SMA) and calponin,
the specific contractile markers of VSMCs. Silencing of NF90/110 in
HAVSMCs abolished AGE-induced VSMC phenotypic switching
(Fig. 3a). Further, NF90/110 knockdown counteracted DM-induced
VSMC phenotypic switching in diabetic atherosclerotic mice (Fig. 3b).
Apart from VSMC phenotypic switch, AGEs enhanced vascular calcifi-
cation by promoting VSMC apoptosis and MV release[2,11]. Tunnel
staining and flow cytometry assays detected that AGE incubation sig-
nificantly accelerated the apoptosis of HAVSMCs, but the silencing of
NF90/110 abolished the effect of AGE (Fig. 3c, d). Meanwhile, the
silencing of NF90/110 reversed the AGE-induced elevation of the
proapoptotic proteins Bax and cleaved-caspase 3, and reduced the
antiapoptotic protein Bax2 in HAVSMCs (Fig. 3e). Consistent with the
results of cultured HAVSMCs, DM-induced cell apoptosis was sig-
nificantly decreased after NF90/110 knockout in atherosclerotic pla-
ques (Fig. 3f). Further, we assayed the levels of MVs in cultured
HAVSMCs. Compared with the AGE group, NF90/110 silencing evi-
dently prevented the AGE-induced increase of MVs (Fig. 3g). These
results indicated that NF90/110 activation accelerates diabetic ather-
osclerotic calcification by mediating AGE-induced VSMC phenotypic
switch, apoptosis, and MVs release.

The AGE-RAGE axis has been reported to mediate multiple sig-
naling pathways that accelerate vascular calcification by promoting
osteoblast-like differentiation and apoptosis in VSMCs, including the
MAPKs pathway, PI3K-AKT pathway, TGF-β-Smads pathway, ROS-NF-
κB pathway and JAK-STAT1 pathway[1,27]. Our results found that AGEs
markedly induced the activation of phosphorylated-p38 (p-p38), p-
ERK1/2, p-AKT, p-Smad1/5, p-NF-κB, and p-STAT1 in cultured
HAVSMCs, and the silencing of NF90/110 attenuated the effect of AGEs
on the phosphorylation of these proteins (Supplementary Fig. 4a).
NF90/110 overexpression was performed using lentivirus-NF90/110
transduction in HAVSMCs, and then treated with AGEs. We found that
overexpression of NF90/110 and AGE treatment much higher raised
the levels of p-p38, p-ERK1/2, p-AKT, p-Smad1/5, p-NF-κB, and p-STAT1
compared to those treated with AGEs alone. Further, we observed that
the interaction of AGE and RAGE that was inhibited by the anti-RAGE

antibody obviously reversed the effect of overexpression of NF90/110
(Supplementary Fig. 4b). These data suggested that NF90/110 accel-
erated osteoblast-like differentiation and apoptosis in VSMCs by acti-
vating AGE-mediated signaling pathways.

## NF90 mediates AGE-induced ubiquitination degradation of
## AGER1 in VSMCs

To examine the mechanism underlying AGER1 modulation by NF90/
110, we used Flag-labeled plasmas of overexpressing NF90 and NF110
to transfect HAVSMCs. Western blot results revealed that AGER1 pro-
tein was downregulated by NF90 overexpression but not NF110, and
RT-qPCR results showed that AGER1 mRNA was not affected by both
NF90 and 110, suggesting that the downregulation of AGER1 protein
level by NF90 is achieved through protein degradation in VSMC
(Fig. 4a, b). Proteins can be degraded via autophagy, proteasome, and
lysosomal pathways. We used the autophagy inhibitor,
3-methyladenine (3-MA), lysosomal inhibitor, chloroquine (CQ), and
proteasome inhibitor, Z-Leu-Leu-Leu-al (MG132), to pre-treat VSMCs
for 24 h, and found that NF90 overexpression-induced AGER1 degra-
dation was abolished by MG132 but not 3-MA or CQ, suggesting that
NF90 mediates proteasomal degradation of AGER1 (Fig. 4c–e).

NF90 and NF110 isoform regulate different steps of gene
expression[15]. For mechanism research, we constructed NF90 siRNA
and NF110 siRNA to transfect HAVSMCs and detect the efficiency of
siRNA by western blotting (Fig. 4f, g). The application of the cyclo-
heximide (CHX) blocking protein translation showed that AGER1 levels
were markedly degraded after CHX treatment. However, NF90
knockdown deferred endogenous AGER1 degradation (Fig. 4h). CHX
and MG132 were used to incubate together HAVSMCs. As shown in
Fig. 4i, proteasomal inhibition reversed the degradation of endogen-
ous AGER1, but NF90 siRNA abolished the effect of proteasomal inhi-
bition. Further, Co-IP and western blot analyses confirmed that NF90
knockdown decreased AGER1 ubiquitination levels in HAVSMCs with
or without AGE treatment (Fig. 4j, k). Moreover, in HEK293T cells
transfected with Flag-labeled NF90, Myc-labeled AGER1, and HA-
labeled ubiquitin, Co-IP assay showed that NF90 overexpression dra-
matically increased Myc-AGER1 ubiquitination (Fig. 4l). These results
demonstrate that NF90 mediates AGE-induced AGER1 protein degra-
dation through the ubiquitin-proteasome pathway in VSMCs.

## NF90 promotes AGEs-induced ubiquitination of AGER1 by
## up-regulation of FBXW7 in VSMCs

NF90 is dsRNA binding proteins that regulates gene expression by
stabilizing mRNAs[28]. To investigate the mechanism of NF90-mediated
AGE-induced ubiquitination of AGER1, RNA immunoprecipitation
sequencing was performed to screen the mRNAs of ubiquitin enzymes
that interact with NF90 in HAVSMC. The results showed that AGE-
induced the mRNAs of seven ubiquitin enzymes that interact with
NF90 when compared to that in BSA, including HECT domain-
containing ubiquitin ligase 1 (HECTD1), ariadne RBR E3 ubiquitin pro-
tein ligase 1 (ARIH1), WWE domain-containing protein 1 (HUWE1),
ARIH2, ubiquitin protein ligase E3 component n-recognin 5 (UBR5),

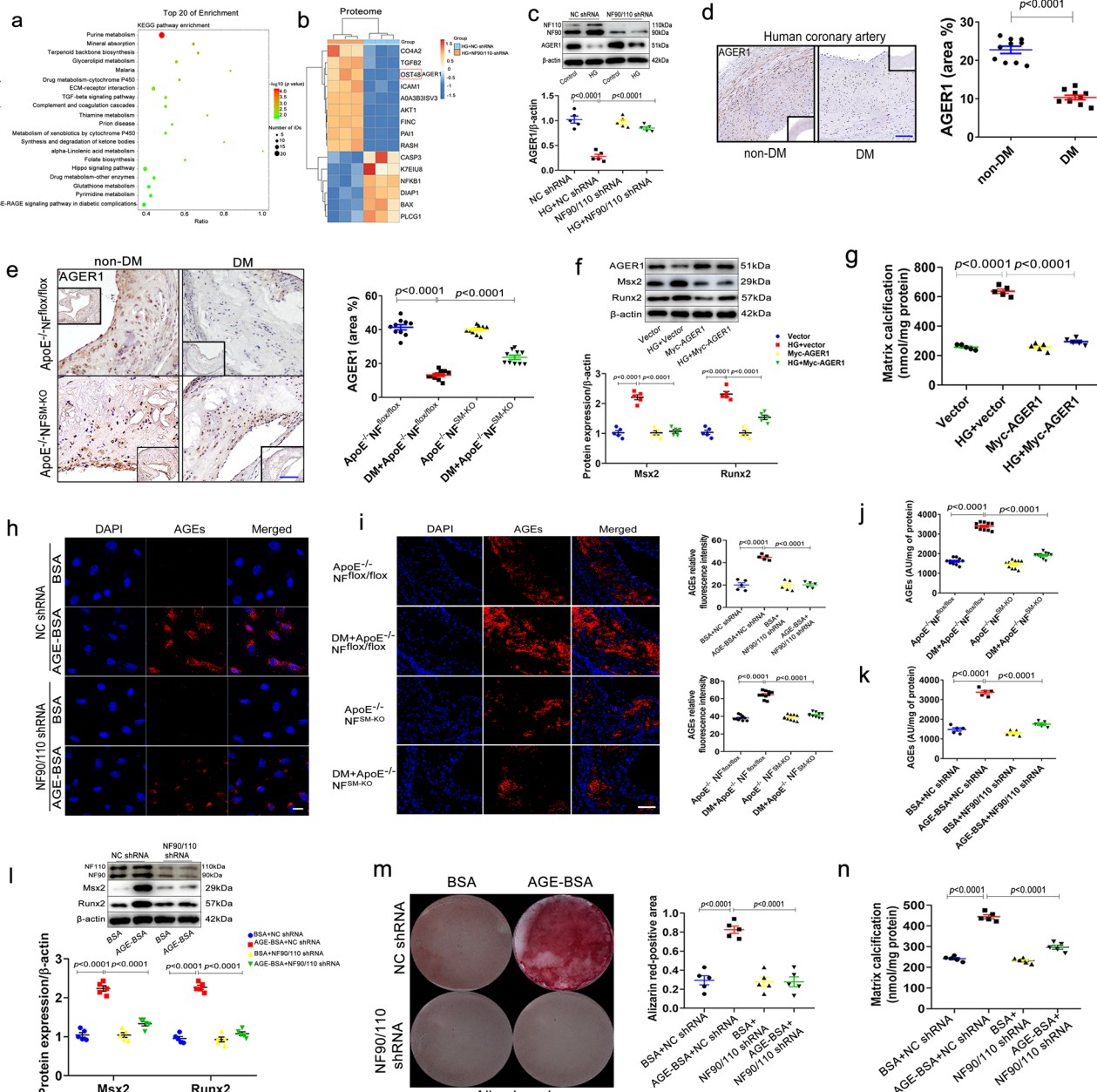

**Fig. 2 | VSMC NF90/110 downregulates AGER1 protein level and increases AGEs accumulation. a, b** Proteomic sequencing was performed in high glucose (HG; 27.5 mM)-stimulated HAVSMCs with or without silencing of NF90/110 (n = 3 per group). **a** Bubble diagram showed the top 20 signaling pathways of KEGG enrichment analysis. **b** Heatmap showing the upregulated and downregulated proteins involved in AGE-RAGE signaling pathway (n = 3 per group). **c** Western blot of AGER1 in HAVSMCs that were transfected with shRNA for NF90/110 silencing and cultured in osteogenic medium with HG (27.5 mM) for four days (n = 5 per group).
**d** Immunostaining of AGER1 in human coronary arteries with or without diabetes mellitus (DM; Scale bar: 100 μm, n = 10 per group). **e** Immunostaining and quantification of AGER1 in aortic roots of $ApoE^{-/-}NF^{flox/flox}$ and $ApoE^{-/-}NF^{SM-KO}$ mice with or without DM (Scale bar: 50 μm, n = 10 per group). **f, g** HAVSMCs were transfected with overexpressing AGER1 plasmids and then incubated in an osteogenic medium with HG (27.5 mM) for four days (n = 5 per group). **f** Western blot and quantification of Msx2 and Runx2 expressions. **g** Total matrix calcium contents of HAVSMCs were

quantified using the Calcium Assay Kit. **h** Immunofluorescent staining of AGEs (red) in HAVSMCs (scale bar: 10 μm, n = 5 per group) and **i** aortic roots of $ApoE^{-/-}NF^{flox/flox}$ and $ApoE^{-/-}NF^{SM-KO}$ mice with or without DM (Scale bar: 50 μm, n = 10 per group). Nuclei were stained by DAPI (blue). BSA: bovine serum albumin. **j** Serum AGE assay in $ApoE^{-/-}NF^{flox/flox}$ and $ApoE^{-/-}NF^{SM-KO}$ mice with or without DM (n = 10 per group). **k–n** HAVSMCs were transfected with shRNA for NF90/110 silencing and cultured in osteogenic medium with AGE (200 μg/ml; n = 5 per group). **k** The cellular supernatant AGEs assay was performed (n = 5 per group). **l** Western blot analyses and quantification of Msx2 and Runx2 expressions. **m** Alizarin-red staining of HAVSMCs incubated with AGEs for 21 days. **n** Matrix calcium content in HAVSMCs were measured. Data were presented as mean ± SEM. Two-tailed Student unpaired $t$-test for (**d**). One-way ANOVA followed by Tukey's post-test analysis for (**c**) and (**e–n**). $p$-values were adjusted for comparisons of multiple means. Source data are provided as a Source Data file.

F-box and WD repeat domain-containing 7 (FBXW7), and HECT and RLD domain-containing E3 ubiquitin protein ligase 4 (HERC4) (Fig. 5a). RT-qPCR assay found that NF90 silencing decreased the mRNA levels of seven ubiquitin enzymes, with the mRNA levels of HECTD1, ARIH1,

HUWE1, and FBXW7 being more significantly reduced (Supplementary Fig. 5a). Further, we confirmed that the combination of FBXW7 and AGER1 was only enhanced after AGE treatment by using Co-IP and western blot analyses (Supplementary Fig. 5b). Immunofluorescence

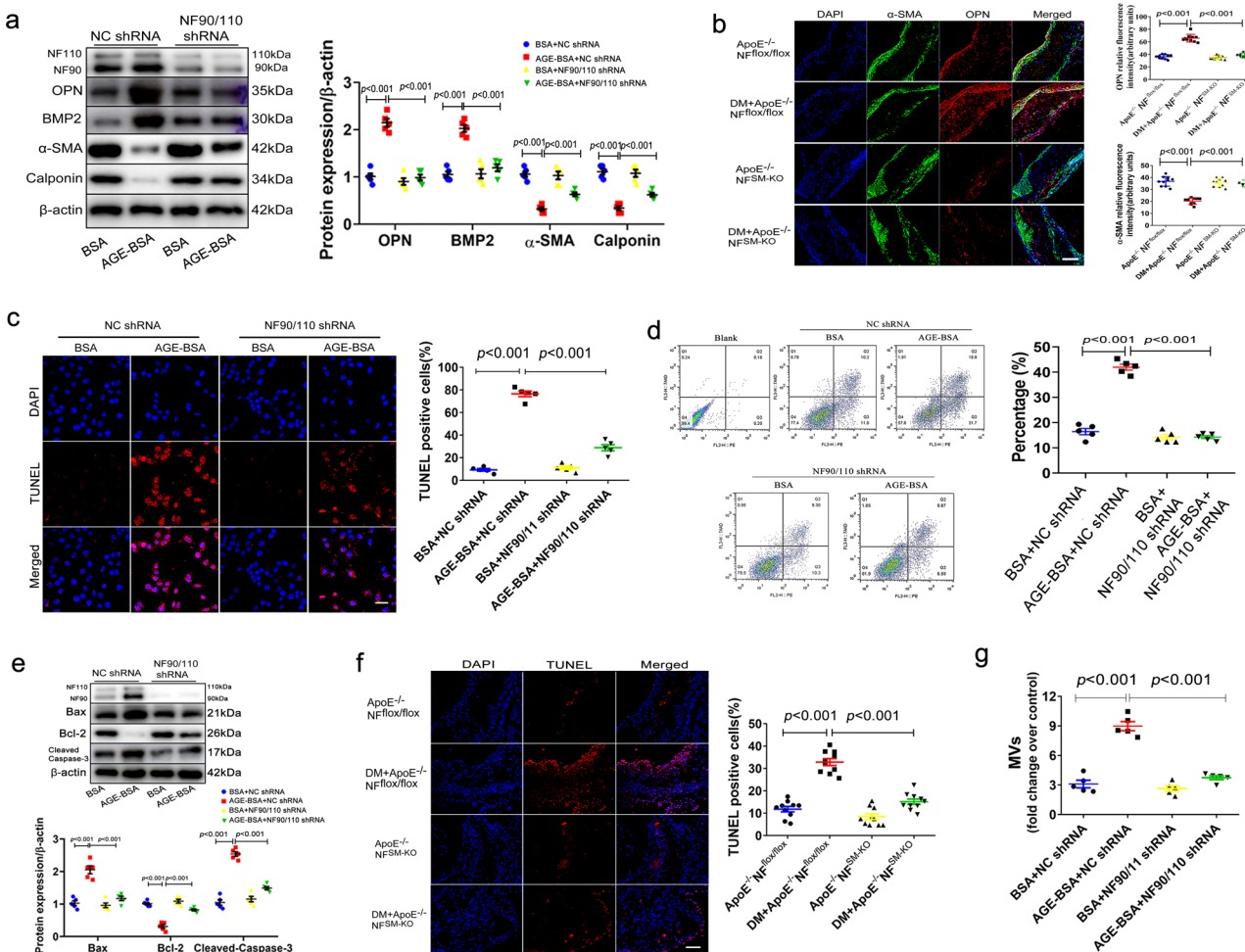

**Fig. 3 | VSMC NF90/110 promotes AGEs-mediated VSMC phenotypic switch, apoptosis, and MVs release. a** The HAVSMCs were pre-infected with NF90/110 shRNA for NF90/110 silencing and then incubated in osteogenic medium with AGE-BSA (200 µg/ml) for 24 h. Representative Western blot bands of OPN, BMP2, α-SMA and Calponin expressions and quantification (n = 5 per group). BSA: bovine serum albumin. **b** Immunofluorescence double staining of OPN (red) and α-SMA (green) in atherosclerotic lesion of $ApoE^{-/-}$ $NF^{flox/flox}$ and $ApoE^{-/-}NF^{SM-KO}$ mice with or without diabetes mellitus (DM) and quantification. Blue DAPI shows positions of nuclei (n = 10 per group, Scale bar: 50 µm). **c–e** The HAVSMCs were pre-infected with NF90/110 shRNA for NF90/110 silencing and then incubated in osteogenic medium with AGEs (200 µg/ml) for 24 h (n = 5 per group). **c** VSMCs apoptosis were evaluated by TUNEL staining. Red fluorescence indicates apoptotic cells and blue

DAPI shows positions of nuclei (Scale bar: 20 µm) and their quantitative analysis. **d** The apoptotic rate was evaluated by PE/Annexin V double staining and their quantification. **e** Representative immunoblots of VSMCs with anti- Bax, Bcl-2 and cleaved-caspase 3 antibodies were shown and quantitative analysis. **f** TUNEL staining in atherosclerotic lesion of $ApoE^{-/-}NF^{flox/flox}$ and $ApoE^{-/-}NF^{SM-KO}$ mice with or without DM and quantification. Red fluorescence represents apoptotic cells and blue DAPI shows positions of nuclei (n = 10 per group, Scale bar: 50 µm). **g** MVs were isolated from supernatant of cultured medium and measured by protein content (n = 5 per group). Data were presented as mean ± SEM. One-way ANOVA followed by Tukey's post-test analysis for (**a–g**). *p*-values were adjusted for comparisons of multiple means. Source data are provided as a Source Data file.

staining displayed that AGE-induced more co-localization of NF90 and FBXW7 in HAVSMCs (Supplementary Fig. 5c). Further, studies have reported that only FBXW7 among them is positively correlated with the homeostasis of glycolipid metabolism and pathogenesis of DM[29–31]. So, we identified the FBXW7 as a ubiquitin enzyme mediating NF90-induced ubiquitination degradation of AGER1. To observe the time node of FBXW7 expression changes regulated by HG and AGEs, immunoblotting assays were performed to detect that FBXW7 increased significantly on days 4 and 5 of HG stimulation, but 1day of AGEs (Supplementary Fig. 6a, b). Next, we found that FBXW7 siRNA administration abolished the downregulation of AGER1 induced by AGE in HAVSMCs (Fig. 5b). Further, HAVSMCs were co-transfected with Flag-NF90 and FBXW7 siRNA under AGE incubation, we found that NF90 overexpression decreased AGER1 protein level, but FBXW7 siRNA inhibited this effect of NF90 overexpression (Fig. 5c). Immunofluorescence staining of HAVSMCs revealed that both FBXW7 and AGER1 were primarily co-localized in the cytoplasm, and AGE

treatment significantly increased the expression of FBXW7 but weakened that of AGER1. Only NF90, but not NF110 knockdown, reversed the effects of AGE treatment (Fig. 5d).

Furthermore, Co-IP and immunoblotting analyses showed the interaction between FBXW7 and AGER1, and AGE increased FBXW7 and AGER1 binding significantly in HAVSMCs (Fig. 5e). In addition, the CHX chase assay revealed that FBXW7 knockdown increased the half-life of endogenous AGER1 protein in HAVSMCs (Fig. 5f). CHX and MG132 were used to show that silencing FBXW7 significantly decreased AGER1 degradation (Fig. 5g). Co-IP analyses revealed that FBXW7 silencing decreased ubiquitin-AGER1 levels in HAVSMCs with or without AGE treatment (Fig. 5h, i). In HEK293T cells, we found that FBXW7 silencing weakened the ubiquitination of Myc-tagged AGER1 (Fig. 5j). Next, we confirmed that overexpression of NF90 increased the ubiquitination levels of AGER1 (Fig. 5k, l). Moreover, silencing of FBXW7 attenuated AGER1 ubiquitination induced by NF90 overexpression in HAVSMCs and HEK293T cells (Fig. 5k, l). These results reveal that NF90 promotes

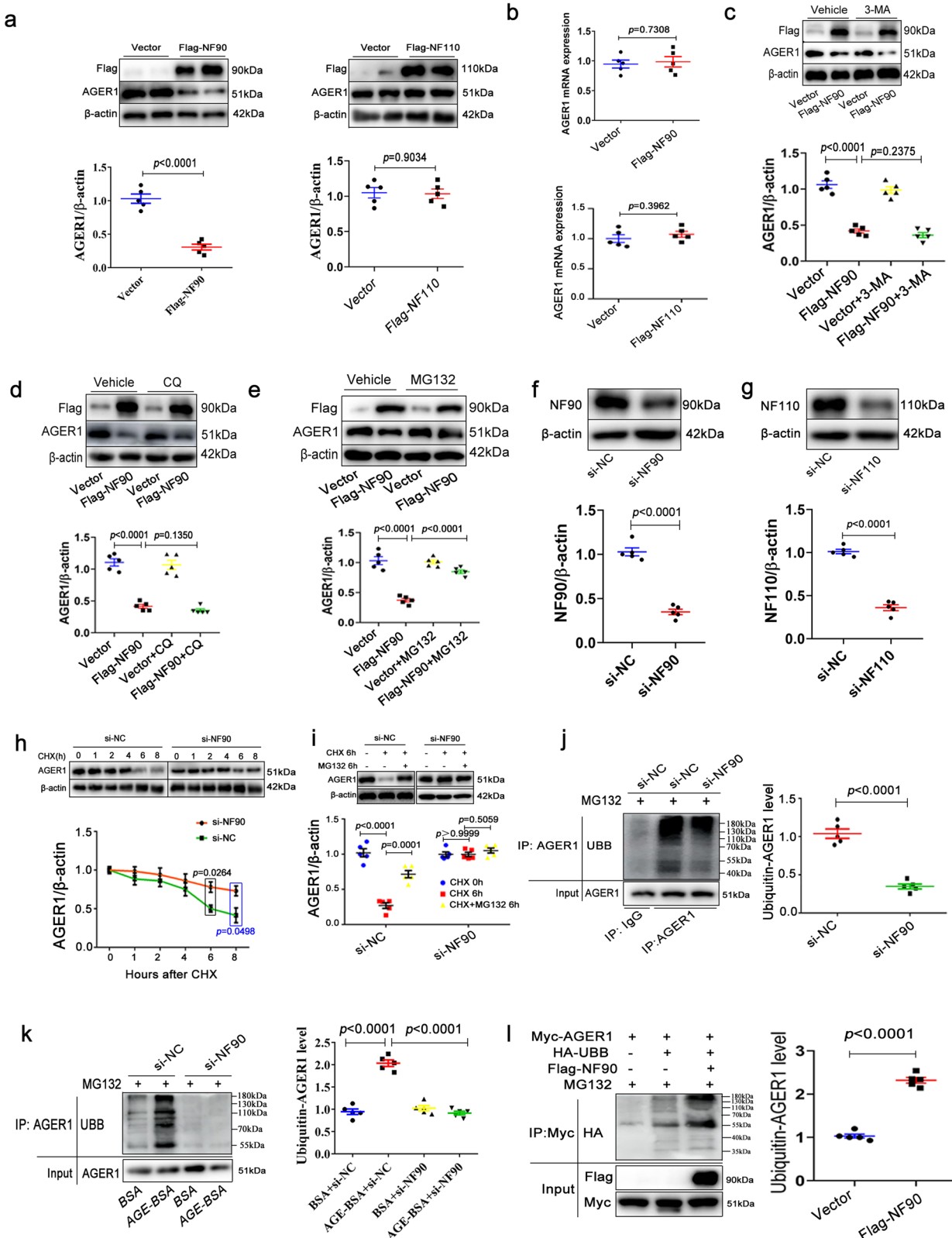

AGE-induced ubiquitination of AGER1 by upregulating FBXW7 in VSMC.

## NF90 up-regulates FBXW7 expression by enhancing the FBXW7 mRNA stability in VSMCs

NF90 maintains mRNA stability by binding their AU-rich element (ARE) of the 3′-untranslated region (UTR), thus regulating gene expression[32–34]. Indeed, there are two AREs in the 3′-UTR of FBXW7, which are the possible sites of binding for NF90 (Fig. 6a). RNA immunoprecipitation assays showed that AGE treatment increased the binding of NF90 to the AREs within the 3′-UTR of FBXW7 (Fig. 6b). Furthermore, the role of NF90 in FBXW7 mRNA stability was evaluated using actinomycin D in cultured HAVSMCs. We found that NF90 knockdown promoted the degradation of FBXW7 mRNA than that in

**Fig. 4 | VSMC NF90 downregulates AGER1 protein through the ubiquitin-proteasome pathway. a, b** Western blot and RT-qPCR, respectively, of AGER1 in HAVSMCs were transfected with plasmas expressing NF90 and NF110 (Flag-NF90 and Flag-NF110; n = 5 per group). **c–e** Western blot of AGER1 in HAVSMCs transfected with or without Flag-NF90 and then pre-treated by 3-methyladenine (3-MA; 10 mM), chloroquine (CQ; 10 μM) and MG132 (10 μM) for 48 h respectively (n = 5 per group). **f, g** The HAVSMCs were pre-infected with NF90 siRNA (si-NF90) and NF110 siRNA (si-NF110) and the efficiency was tested by western blot (n = 5 per group). **h** HAVSMCs were transfected with NF90 siRNA or NC siRNA, then were treated with CHX (50 μg/ml) for 0, 1, 2, 4, 6 and 8 h followed by immunoblotting with the AGER1 antibody (n = 5 per group). **i** HAVSMCs were treated with cyclo-heximide (CHX; 50 μg/ml) or MG132 (10 μM) for 0 h or 6 h and blotted for AGER1 and β-actin (n = 5 per group). **j** HAVSMCs were transfected with NF90 siRNA or NC siRNA and pre-treated with MG132 (10 μM) for 24 h, and then harvested for co-immunoprecipitation (Co-IP) and immunoblotting (n = 5 per group) UBB: ubiquitination. **k** The HAVSMCs were pre-infected with NF90 siRNA and then incubated in osteogenic medium with AGE (200 μg/ml) for 24 h and MG132 (10 μM) for 24 h. Co-IP and immunoblotting were performed to test ubiquitin-AGER1 level (n = 5 per group). **l** HEK293T cells were transfected with the indicated plasmids for 48 h and then treated with MG132 (10 μM) for 24 h, followed by Co-IP with indicated beads and then immunoblotting (n = 5 per group). Data were presented as mean ± SEM. Two-tailed Student unpaired *t*-test for (**a, b, f–h, j**, and **l**). One-way ANOVA followed by Tukey's post-test analysis for (**c–e, l**, and **k**). *p*-values were adjusted for comparisons of multiple means. Source data are provided as a Source Data file.

NC shRNA (Fig. 6c). Immunoblotting displayed that the NF90 siRNA but not NF110 siRNA decreased the FBXW7 level in HAVSMCs (Fig. 6d, e). Immunohistochemical staining showed that hyperglycemia increased expression of FBXW7 in the human and mice arteries with diabetes, and FBXW7 expression was inhibited in DM+*ApoE*$^{-/-}$*NF*$^{SM-KO}$ mice than DM+*ApoE*$^{-/-}$*NF*$^{flox/flox}$ mice (Fig. 6f, g). Further, NF90 knockdown significantly reduced AGE-induced elevation of FBXW7 in HAVSMC (Fig. 6h). Interestingly, immunofluorescence revealed that FBXW7 localized mainly to the cytoplasm, while NF90 and NF110 localized to the nucleus (Fig. 6i). However, AGE significantly increased the cytoplasmic shift of NF90 and promoted the co-localization of NF90 and FBXW7 in the cytoplasm. Knockdown of NF90, but not NF110, disrupted this co-localization of NF90 and FBXW7 in HAVSMC stimulated with AGE (Fig. 6i). These results support that NF90 enhances FBXW7 mRNA stability and expression by binding to the AREs within the 3′-UTR.

In sum, hyperglycemia accelerates the production of AGEs in diabetes, AGEs increase the expression of NF90 and mediate its cytoplasmic translocation in VSMC. Cytoplasmic NF90 enhances bound with the 3′-UTR of FBXW7 that maintained FBXW7 mRNA stability and increased its protein expression. As an E3 ubiquitin ligase, FBXW7 binds to AGER1 and mediates its ubiquitination and degradation. Ubiquitin degradation of AGER1 in turn leads to enhanced more AGEs accumulation (Fig. 6j).

### NF90-FBXW7-AGER1 axis promotes AGEs-induced diabetic atherosclerotic calcification

To confirm that the NF90-FBXW7-AGER1 axis mediates AGE-induced atherosclerotic calcification, we investigated whether inhibition of the NF90-FBXW7-AGER1 axis abolished the AGE-induced phenotypic transition, apoptosis, and MVs content of VSMCs. Silencing of NF90 and FBXW7, or overexpression AGER1 in HAVSMCs abolished AGE-induced smooth muscle cells phenotypic switch of contractility (decreased α-SMA and calponin) to osteogenic type (increased OPN, BMP2 and Runx2, Fig. 7a). Further, the AGE-induced elevation of alkaline phosphatase (ALP) activity in HAVSMCs was inhibited by silencing of NF90 and FBXW7, or overexpression AGER1 (Fig. 7b). Immunoblotting showed that silencing of NF90 and FBXW7 or over-expression of AGER1 in HAVSMCs significantly decreased the AGE-induced elevation of Bax, cleaved-caspase 3 and Runx2, but increased the expression of Bcl-2 (Fig. 7c). NF90 or FBXW7 silencing in HAVSMCs inhibited the AGE-induced increase of MV. Moreover, overexpression of AGER1 displayed a similar result (Fig. 7d).

We conducted a FBXW7 rescue experiment by using an adenovirue-FBXW7 vector (Ad-FBXW7) to overexpress FBXW7 in HAVSMCs or an adeno-associated virus- FBXW7 vector (AAV-FBXW7) containing the SM22α promotor region to target VSMCs to overexpress FBXW7 in DM+*ApoE*$^{-/-}$*NF*$^{SM-KO}$ mice. Western blot assay showed that AAV-FBXW7 was only expressed specifically in VSMCs not macrophages and endotheliocytes from AAV-FBXW7 mice (Supplementary Fig 7a). We found that Ad-FBXW7 inhibited AGER1 activation and significantly reversed OPN, Bax and Runx2 expressions and ALP activity in NF90 knockdown HAVSMCs after AGEs incubation (Fig. 7e, f). Similar results were observed in the arteries of DM+*ApoE*$^{-/-}$*NF*$^{SM-KO}$ and DM+*ApoE*$^{-/-}$*NF*$^{SM-KO}$ + AAV-FBXW7 mice using immunohistochemical staining (Fig. 7g). In summary, these results revealed that the NF90-FBXW7-AGER1 axis plays a crucial role in AGEs-induced diabetic atherosclerotic calcification.

## Discussion

DM is a group of metabolic disorders characterized by chronic hyperglycemia that leads to vascular complications[35]. Hyperglycemia is strongly associated with atherosclerosis through several pathological pathways, such as oxidative stress, AGEs generation, protein kinase C (PKC) signaling, chronic inflammation, circulating non-coding RNAs, and epigenetic modification[36]. Hyperglycemia-mediated the calcification of atherosclerotic plaques is an important pathological feature of diabetes[22]. Several mechanisms have been raised to state how DM deteriorates vascular calcification, including inflammation, oxidative stress, AGEs, and inorganic phosphate[3–5]. Among them, the metabolic imbalance of AGE plays a central role in the pathophysiological processes that lead to the development of atherosclerotic calcification in diabetes[37]. In this study, we demonstrated that AGEs-mediated NF90 activation causes ubiquitination and proteasome degradation of AGER1 by enhancing the mRNA stability of FBXW7, which in turn leads to more accumulation of AGEs in VSMCs. Excessive accumulation of AGEs enhances the interaction with RAGE, and promotes VSMC phenotypic switching, apoptosis, and MVs release by activating multiple signaling pathways downstream of AGE-RAGE, and accelerates the diabetic atherosclerotic calcification (Fig. 8a). In contrast, knockout of NF90 leads to downregulation of FBXW7 expression, elevation of AGER1 protein level, and reduction of AGEs which further alleviates diabetic atherosclerotic calcification (Fig. 8b). Therefore, we conclude that inhibition of VSMC NF90 may be a potential therapeutic target for diabetic atherosclerotic calcification.

The mechanisms have been raised to state how AGEs accelerate atherosclerotic calcification in diabetes. On the one hand, chronically elevated hyperglycemia markedly accelerated the AGEs production[6], on the other hand, hyperglycemia specifically impaired the clearance of AGEs[38]. Excessive accumulation of AGEs activates RAGE, and accelerates atherosclerotic calcification by modifying phenotype switch of VSMC from contractile to osteoblastic phenotype, apoptosis and MV release[11,26]. We revealed that VSMC NF90/110 is involved in the pathogenesis of AGEs-mediated diabetic atherosclerotic calcification. Furthermore, VSMC-specific deletion of NF90/110 decreased AGEs accumulation to attenuate diabetic atherosclerotic calcification, identifying the causative function of VSMC NF90/110 in this disorder. Further, we provide evidence suggesting that the NF90-FBXW7- AGER1 axis mediates AGE-induced VSMC osteogenic differentiation and apoptosis and increases the release of MV, which cause atherosclerotic calcification in diabetes. The loss of NF90 can block VSMC osteogenic differentiation, apoptosis, MV secretion, and subsequent atherosclerotic calcification mediated by AGEs in diabetes.

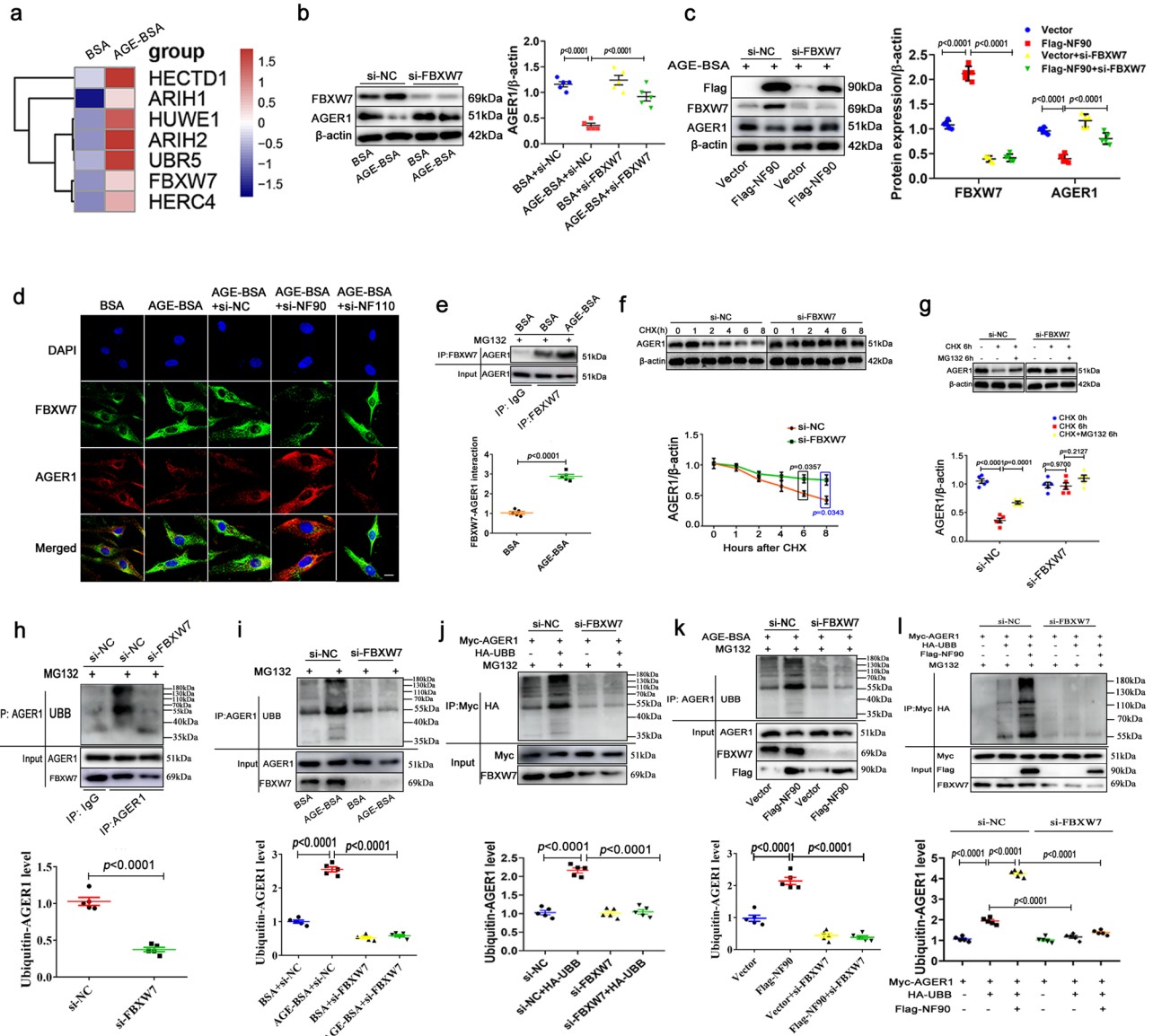

**Fig. 5 | VSMC NF90 mediates AGEs-induced AGER1 ubiquitination-proteasomal degradation by enhancing FBXW7 expression. a** After HAVSMCs were incubated with AGE-BSA for 24 h, RNA immunoprecipitation-seq test was performed and heatmap showing the differential mRNA levels involved in ubiquitination upon NF90 immunoprecipitation. BSA: bovine serum albumin. **b** The HAVSMCs were pre-infected with FBXW7 siRNA (si-FBXW7) and then incubated with AGEs (200 μg/ml) for 24 h. Representative immunoblots and quantitation of FBXW7 and AGER1 proteins (n = 5 per group). **c** The HAVSMCs were incubated with Flag-NF90 and FBXW7 siRNA as well as AGEs (200 μg/ml) for 24 h. Representative immunoblots and quantitation of FBXW7 and AGER1 proteins (n = 5 per group). **d** Immunofluorescence double staining of FBXW7 (green) and AGER1 (red) in HAVSMCs. Blue DAPI shows positions of nuclei (Scale bar: 10 μm; n = 5 per group). **e** Co-IP showed the interaction of FBXW7 and AGER1 in HAVSMCs with or without AGE (200 μg/ml) and MG132 (10 μM) for 24 h (n = 5 per group). **f** FBXW7 knockdown extends the protein half-life of AGER1 in HAVSMCs with 50 μg/ml cycloheximide

(CHX) for indicated times (n = 5 per group). **g** Western blot of AGER1 in HAVSMCs that were treated with CHX (50 μg/ml) or CHX+ MG132 (10 μM) for 6 h (n = 5 per group). **h, i** HAVSMCs were transfected with FBXW7 siRNA and pre-treated with MG132 (10 μM) or MG132+AGEs (200 μg/ml) for 24 h. co-immunoprecipitation (Co-IP) and immunoblotting were performed to test ubiquitin-AGER1 level (n = 5 per group). UBB: ubiquitination. **j** HEK293T cells were transfected with FBXW7 siRNA and the indicated plasmids for 48 h, followed by Co-IP and immunoblotting (n = 5 per group). **k** HAVSMCs were transfected with the Flag-NF90 and FBXW7 siRNA and then treated with AGEs (200 μg/ml) and MG132 (10 μM) for 24 h, Co-IP and immunoblotting with indicated antibodies (n = 5 per group). **l** HEK293T cells transfected with FBXW7 siRNA and the indicated plasmids for 48 h, followed by Co-IP and immunoblotting with indicated antibodies (n = 5 per group). Data were presented as mean ± SEM. Two-tailed Student unpaired *t*-test for (**e, f**, and **h**). One-way ANOVA followed by Tukey's post-test analysis for (**b, c, g**, and **i–l**). *p*-values were adjusted for comparisons of multiple means. Source data are provided as a Source Data file.

Epidemiological investigations have shown that cumulative chronic hyperglycemia exposure increases the incidence of cardiovascular disease in diabetes[39]. AGEs, rather than hyperglycemia, were identified to be responsible for vascular complications in diabetes[37]. The abnormal accrual of AGEs in the body is related to input from endogenous and exogenous sources[40]. The endogenous generation of AGEs is markedly increased by hyperglycemia and elevated oxidative stress[41]. Under normal physiological conditions, AGEs will be

removed before they produce additional protein modifications and protein crosslinking. AGER1 plays a major role in the clearance of AGEs through receptor mediated endocytosis[27]. The AGE-AGER1 complex forms clathrin-mediated vesicles by activating the phosphorylation or ubiquitinoylation of AGER1. These vesicles are transported to endosomes which then fused with lysosomes, leading to degradation of AGEs[27]. Clinical studies show significantly decreased expression of AGER1 in patients with diabetes[38,42]. Our findings

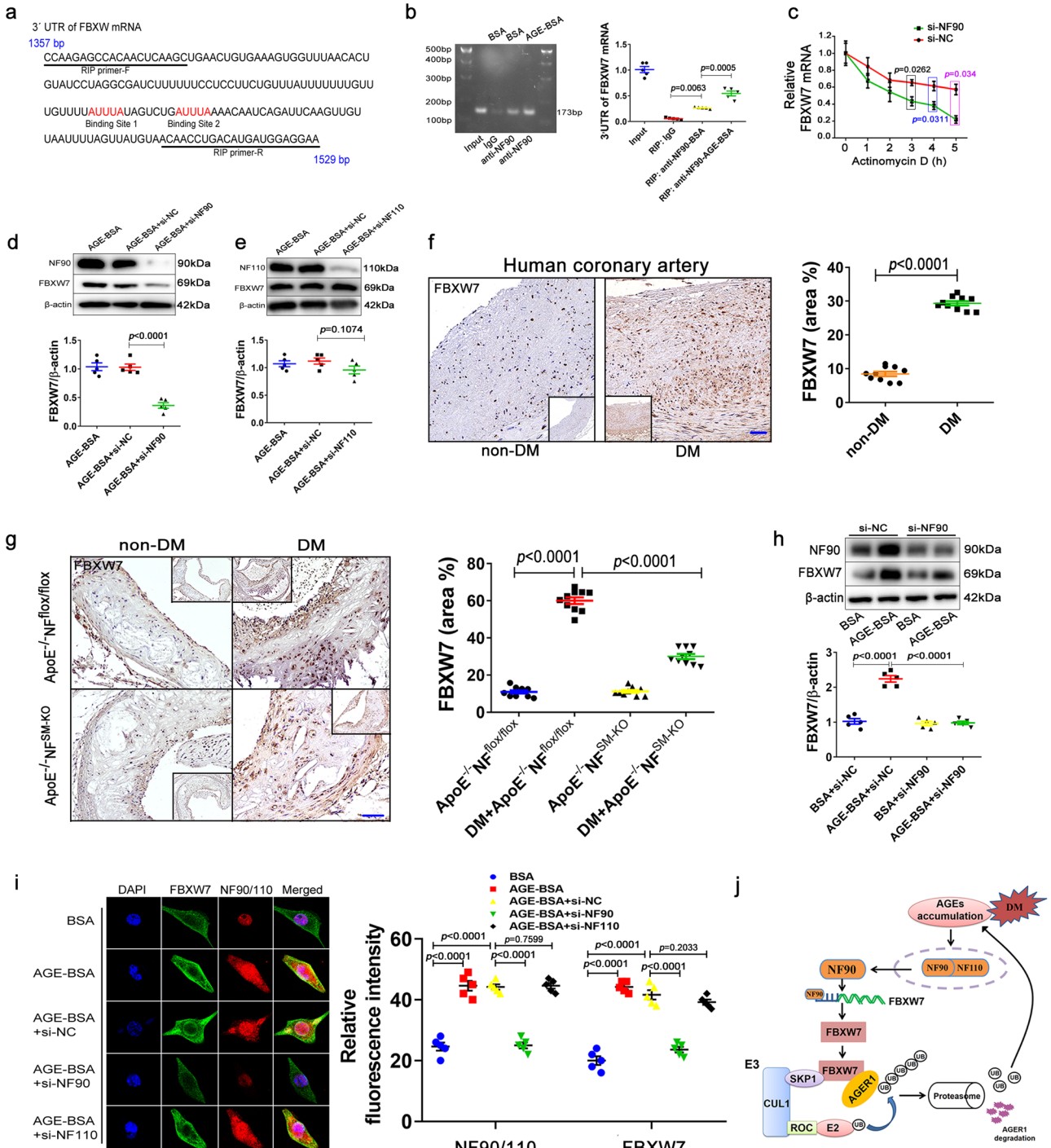

**Fig. 6 | VSMC NF90 mediates AGEs-induced increase in FBXW7 expression by stabilizing FBXW7 mRNA. a** The partial 3′-untranslated regions (UTR) sequence of FBXW7 mRNA from 1357 bp to 1529 bp. The AUUUA motifs are highlighted in red. The primer sequences for RNA immunoprecipitation (RIP) assay are underlined. **b** RNA immunoprecipitation assay was performed with specific primers to identify binding of NF90 to the AUUUA-rich 3′-UTR of FBXW7 mRNA and quantification of NF90 binding (n = 5 per group). BSA: bovine serum albumin. **c** RT-qPCR analysis and quantification of FBXW7 mRNA levels in HAVSMCs that were transfected with NF90 siRNA (si-NF90) and treated with Actinomycin D for 0, 1, 2, 3, 4 and 5 h (n = 5 per group). **d**, **e** Western blot and quantification of NF90 or NF110 and FBXW7 in HAVSMCs that were incubated in osteogenic medium with AGE (200 μg/ml) stimulation for 24 h and transfected with NF90 siRNA and NF110 siRNA (n = 5 per group). **f** Immunostaining and quantification of FBXW7 in human atherosclerotic arteries with or without diabetes mellitus (non-DM; Scale bar: 100 μm, n = 10 per

group). **g** Immunostaining and quantification of FBXW7 in aortic roots of *ApoE*[−/−]*NF*[flox/flox] and *ApoE*[−/−]*NF*[SM-KO] mice with or without DM (Scale bar: 50 μm, n = 10 per group). **h** Western blot and quantification of FBXW7 in HAVSMCs treated with NF90 siRNA and or or AGE (200 μg/ml) (n = 5 per group). **i** Immunofluorescence double staining of FBXW7 (green) and NF90/110 (red) in HAVSMCs that were incubated in osteogenic medium with AGE (200 μg/ml) for 24 h and then pre-infected with NF90 siRNA or NF110 siRNA. Nuclei stained by DAPI (blue), and yellow indicates the co-localization of both proteins (Scale bar: 10 μm, n = 5 per group). **j** Schematic diagram of AGEs-induced NF90/110 promoting AGER1 ubiquitination by elevating FBXW7. Data were presented as mean ± SEM. Two-tailed Student unpaired *t*-test for (**c**–**f**). One-way ANOVA followed by Tukey's post-test analysis for (**b**) and (**g**–**i**). *p*-values were adjusted for comparisons of multiple means. Source data are provided as a Source Data file.

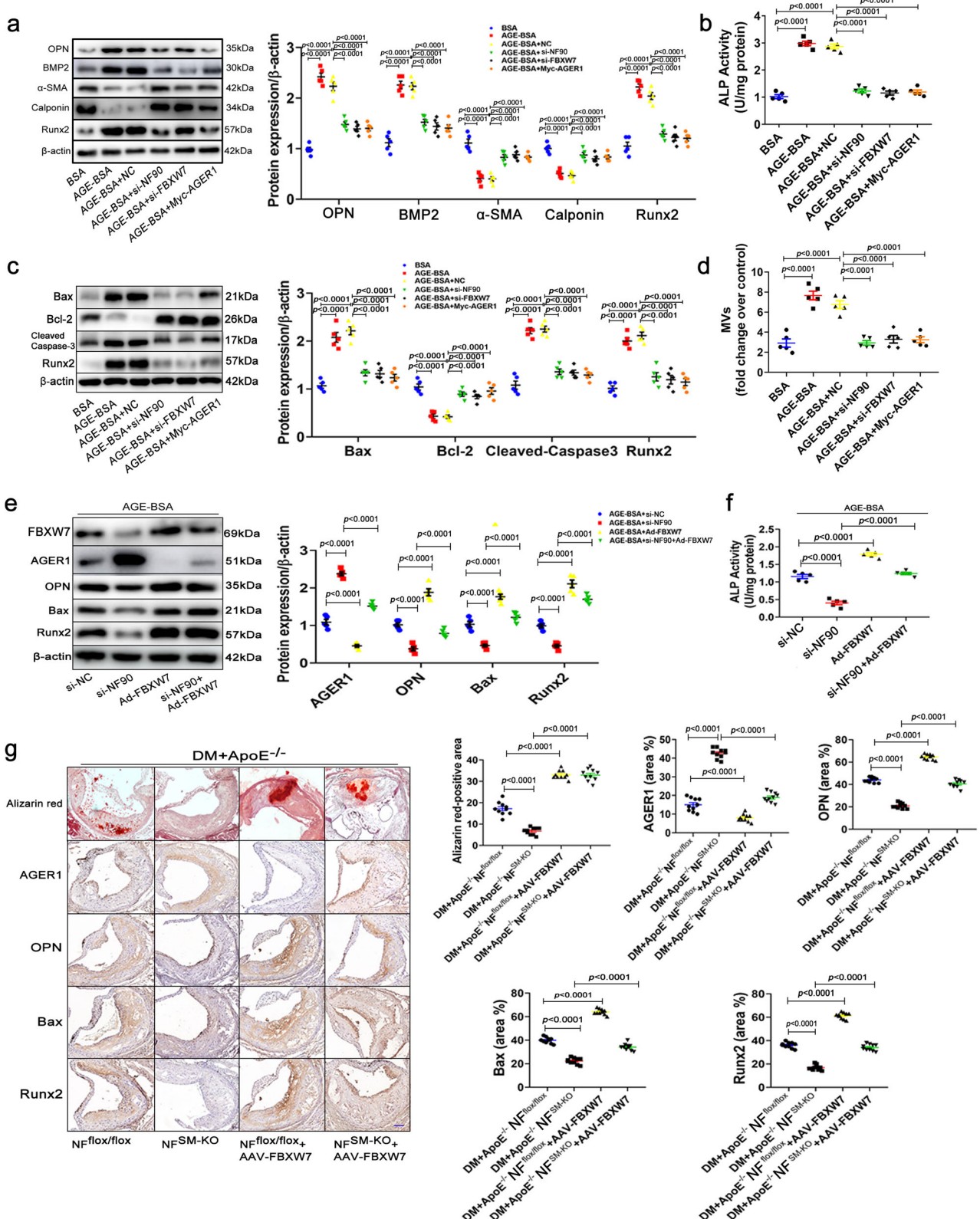

showed that the level of AGER1 in VSMCs was decreased after incubation with AGEs. Although knockdown of NF90 in VSMCs increased AGER1 level and decreased AGEs accumulation. Based on these results, we speculate that NF90 may be associated with the increase of AGEs in VSMC. Proteomic analysis and our studies revealed that NF90 expression was negatively correlated with AGER1 protein but not mRNA levels. Our findings suggest that NF90 promotes AGEs accumulation in VSMC by increasing AGER1 degradation.

NF90 is a dsRNA binding protein and regulates gene expression by stabilizing mRNAs. In the current study, we revealed that AGEs increased the binding of NF90 with the 3′-UTR of FBXW7 mRNA and upregulated its expression. Some studies showed that FBXW7 is an E3

**Fig. 7 | NF90-FBXW7-AGER1 axis promotes diabetic atherosclerotic calcification. a–d** HAVSMCs were transfected with NF90 siRNA (si-NF90), FBXW7 siRNA (si-FBXW7), or Myc-AGER1 plasmid after AGEs (200 μg/ml) incubation for 24 h (n = 5 per group). BSA: bovine serum albumin. **a** Western blotting of OPN, BMP2, α-SMA, Calponin, and Runx2 protein expression. **b** ALP activity was measured in calcified HAVSMCs. **c** Western blotting of Bax, Bcl-2, cleaved-caspase 3, and Runx2 protein expression. **d** Quantification of MV isolated from the supernatant of HAVSMCs culture medium. **e, f** HAVSMCs were transfected with NF90 siRNA or adenovirus expressing FBXW7 vector (Ad-FBXW7) after AGE (200 μg/ml) incubation for 24 h

(n = 5 per group). **e** Western blotting of AGER1, OPN, Bax, and Runx2 protein expression. **f** ALP activity was measured in calcified HAVSMCs. **g** Alizarin-red staining and immunohistochemical staining of AGER1, OPN, Bax, and Runx2 in aortic roots of *ApoE⁻/⁻NF^flox/flox* and *ApoE⁻/⁻NF^SM-KO* diabetic mice with or without injection of FBXW7 adenovirus associated virus vector (AAV-FBXW7) (Scale bar: 50 μm, n = 10 per group). Data were presented as mean ± SEM. One-way ANOVA followed by Tukey's post-test analysis for (**a–g**). *p*-values were adjusted for comparisons of multiple means. Source data are provided as a Source Data file.

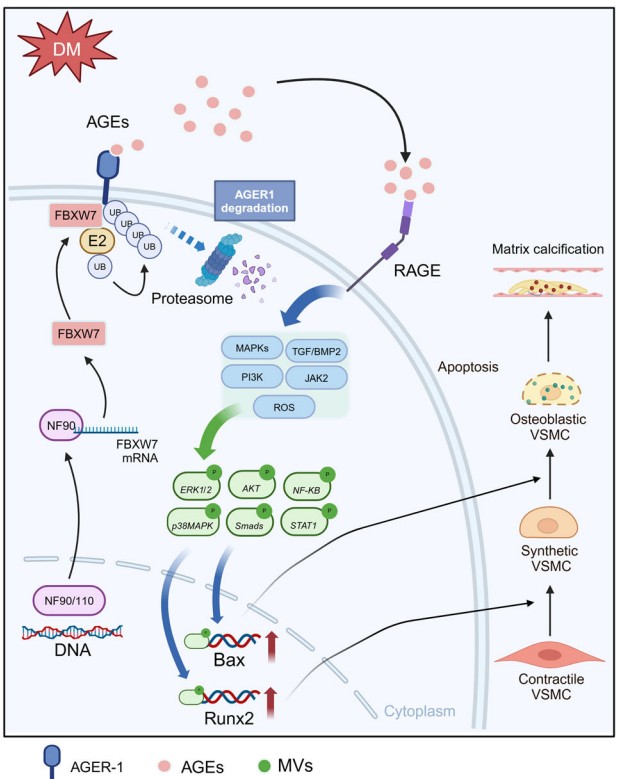
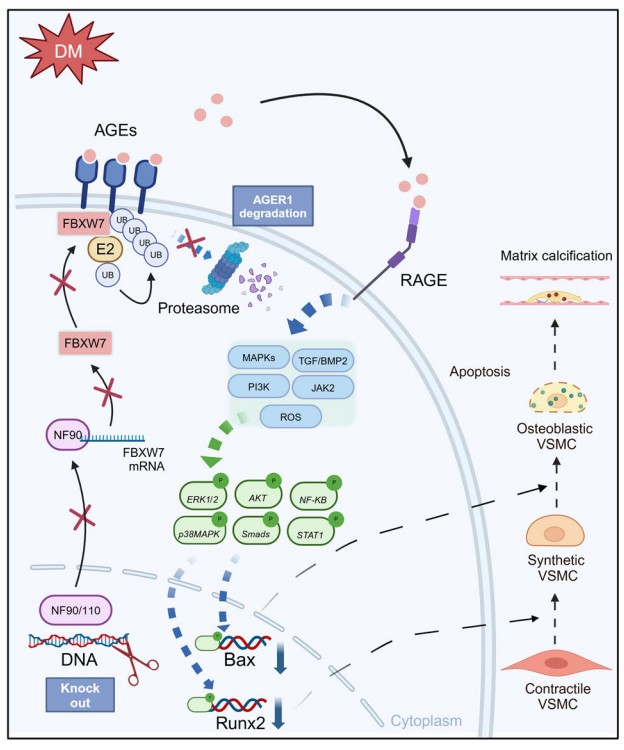

**Fig. 8 | Schematic diagram of NF90-mediated AGEs accumulation and the subsequent acceleration of atherosclerotic calcification in DM. a** In the NF90-sufficient diabetic condition, extracellular levels of AGE are increased due to reduced levels of AGER1, of which degradation is increased via the activated FBXW7/proteasome pathway, and VSMC-mediated matrix calcification is increased via the increased signaling of AGE-RAGE axis. **b** In the NF90-deficient diabetic condition, extracellular levels of AGE are decreased due to increased levels of AGER1, of which degradation is decreased via the inhibited FBXW7/proteasome pathway, and VSMC-mediated matrix calcification is decreased via the reduced signaling of AGE-RAGE axis.

ubiquitin enzyme, and its expression is positively correlated with an imbalance in glycolipid metabolism, and is associated with diabetes[29,43]. However, other studies showed a beneficial role of FBXW7 in glucose homeostasis[30,44]. In light of this contradiction, our current findings showed that hyperglycemia increased the levels of FBXW7 in human arteries in diabetes, and AGEs significantly promoted the protein interaction between FBXW7 and AGER1 in HAVSMCs. In addition, the CHX chase assay revealed that FBXW7 deficiency increased the half-life of endogenous AGER1 protein in HAVSMCs, implying that AGER1 is a potential substrate of FBXW7. Furthermore, we confirmed that the NF90-induced degradation of AGER1 depends on FBXW7 expression. Thus, AGE exaggerate NF90-mediated AGER1 degradation by enhancing FBXW7 mRNA stability under diabetic conditions. This conclusion is supported by earlier reports that AGER1 expression was maintained by AGE restriction in mice and humans[45–47]. In addition, our findings revealed that AGE increased the expression of osteoblastic markers, such as OPN and BMP2, and decreased the level of the contractile markers, α-SMA and calponin, in HAVSMCs. These changes are

dependent on the NF90-FBXW7-AGER1 axis. Moreover, inhibition of the NF90-FBXW7 axis or overexpression of AGER1 resulted in the opposite effects.

Our findings may explain the metabolic memory phenomenon found in diabetic vascular complications. Several studies have indicated that past hyperglycemic exposure continues to cause vascular damage in diabetic patients and animals that are not easily reversed, even with good glycemic control. These persistent adverse effects of hyperglycemia on the development and progression of vascular complications are defined as metabolic memory[48–51]. Although there is increasing evidence that AGEs play an important role in metabolic memory[52,53], the biochemical mechanisms underlying metabolic memory remain unclear. Here, we showed that, AGEs are generated in both extracellular and intracellular conditions and accumulate in the body in chronic hyperglycemia. AGEs induce ubiquitin degradation of AGER1 by activating the NF90-FBXW7 pathway to reduce the clearance of AGEs, which in turn leads to increased AGEs accumulation, that interacts with RAGE resulting in vascular complications in diabetes.

Even with controlled hyperglycemia in diabetic patients, previously accumulated AGEs may continue to promote vascular complications through the NF90-FBWX7-AGER1 axis.

In conclusion, we have identified a novel molecular mechanism underlying diabetes- accelerated atherosclerotic calcification. Chronic hyperglycemia-mediated AGEs generation increases NF90 levels in VSMCs and mediates more AGEs accumulation by augmenting the availability of FBXW7 allowing it to participate in ubiquitination protease degradation of AGER1. The accumulation of AGEs promotes the phenotypic switching of VSMCs to osteoblast-like cells and apoptosis in atherosclerotic plaques. Deletion of NF90 inhibits the phenotypic switching and apoptosis of VSMC by reducing the accumulation of AGEs and alleviates atherosclerotic calcification in diabetes. These findings demonstrate that NF90 inhibition is a potential therapeutic target for diabetes-induced atherosclerotic calcification.

## Methods

### Human coronary artery samples
Based on the considerable evidence suggesting that estrogen modulates cardiovascular physiology and function in both health and disease, and that it could potentially serve as a cardioprotective agent. We considered the need to exclude the effect of gender at the beginning design of study and chose the specimens from male patients and male mice for this current study[54,55].

The specimens were donated by the Shandong Red Cross Society. Atherosclerotic epicardial coronary artery segments were collected from Chinese males with or without DM (n = 10 per group). Ten patients with DM with an average of 40 were included in this study. Ten patients without DM with an average of 39 were included in this study. The detailed information of patients is listed in Supplementary Table 3. All patients gave written informed consent in the study. The experiment protocols were examined and approved by the review committee of Qilu Hospital of Shandong University, Jinan, China (ethics approval No. KYLL-2018(KS)-233).

### Animals
All animal procedures were performed following the protocols approved by the Institutional Animal Care and Use Committee of Shandong University and conformed to the Guide for the Care and Use of Laboratory Animals (NIH publication No. 86-23, revised 1985). All experiments were approved by the Ethics Committee of Shandong University (KYLL-2022(ZM)-039).

NF90/NF110 knockout floxed ($NF^{flox/flox}$) mice under the combined use of CRISPR/Cas9 and Cre/LoxP, SM22a-creERT2 mice, and $ApoE^{-/-}$ mice in a C57BL/6J background were generated by View Solid Biotechnology Inc (Beijing, China). The VSMC conditional NF90/NF110 knockout ($NF^{SM-KO}$) mice were bred from $NF^{flox/flox}$ mice crossed with SM22a-creERT2 mice ($NF^{flox/flox}/Cre^+$). The $ApoE^{-/-}NF^{SM-KO}$ double knockout mice were generated by crossing $ApoE^{-/-}$ mice and $NF^{flox/flox}/Cre^+$ mice. The genotypes were verified by PCR. Littermate $ApoE^{-/-}NF^{flox/flox}/Cre^-$ mice were used as controls. The six-week-old male $ApoE^{-/-}NF^{flox/flox}$ mice and $ApoE^{-/-}NF^{SM-KO}$ mice were injected intraperitoneally with tamoxifen (50 mg/kg) for five consecutive days. On the seventh day after the end of the injections, creERT2 was induced successfully.

All ten-week-old male $ApoE^{-/-}$ background mice were injected intraperitoneally with STZ (Sigma-Aldrich, St. Louis, MO, USA) at a dose of 50 mg/kg for five consecutive days to establish a diabetic model. After two weeks, the twelve-week-old male mice were fed with a high fat diet (HFD; Trophic Animal Feed High-tech Co., Ltd, Nantong, China, No.TP28640; ingredients: 15% fat, 1.25% cholesterol, and 0.5% sodium cholate) for twelve continuous weeks.

These male mice were divided into four groups: non-diabetic $ApoE^{-/-}NF^{flox/flox}$ group (non-DM), diabetic $ApoE^{-/-}NF^{flox/flox}$ group (DM+$ApoE^{-/-}NF^{flox/flox}$), non-DM $ApoE^{-/-}NF^{SM-KO}$ group, DM+$ApoE^{-/-}NF^{SM-KO}$ group (n = 10 per group). Mice were housed in a pathogen-free animal care facility with humidity of 40–60% at a constant temperature (24 °C) and a conventional light/dark cycle (12/12 h) under free conditions. At the end of the experiment, these mice were sacrificed under general anesthesia with sodium pentobarbitone and with efforts to minimize suffering. Mice serum was collected for glucose and lipids analyses, and tissues were harvested for histological examination.

To further evaluate the contribution of the NF90-FBXW7-AGER1 axis in mediating AGE-induced diabetic atherosclerotic VSMC calcification, adeno-associated virus overexpressing FBXW7 (AAV-FBXW7; transcript NM_001177773; including SM22α promotor region to target VSMCs) were injected into twelve-week-old male $ApoE^{-/-}$ background mice with DM (1.5 * 10^11 PFU/200 μl) via the tail vein to generate FBXW7-overexpressed mice. These male mice were divided into four groups: DM+$ApoE^{-/-}NF^{flox/flox}$+vector group, DM+$ApoE^{-/-}NF^{flox/flox}$ + AAV-FBXW7 group, DM+$ApoE^{-/-}NF^{SM-KO}$+vector group and DM+$ApoE^{-/-}NF^{SM-KO}$ + AAV-FBXW7 group (n = 10 per group).

### Plasma biochemistry
The blood of the mice was collected from the left ventricle using heparinized syringes and immediately centrifuged at 4 °C at the time of tissue harvesting. The levels of TG, TC, LDL-C, BG, HDL-C, calcium, and phosphorus were detected using standard enzymatic methods and commercial kits (Roche, Basel, Switzerland).

### AGEs assay
Mice serum and cellular supernatant AGEs concentrations were detected quantitatively using commercial enzyme-linked immuno-sorbent assay kit (Biovision, Waltham, MA, USA) following the manufacturer's instructions. The results were measured using a microplate reader (Bio-Rad, Hercules, CA, USA).

### Immunohistochemistry and immunofluorescence staining
Paraffin-embedded tissue sections of human coronary artery and mice aortic root were deparaffinized and stained with antibodies (1:200) specific for NF90/NF110 (Abcam, ab92355), Msx2 (Abcam, ab223692), Runx2 (Abcam, ab192256), AGER1 (Proteintech, 14916-1-AP), FBXW7 (Abcam, ab109617), OPN (Abcam, ab8448) and Bax (Abcam, ab32503). Samples were incubated with biotin-conjugated secondary antibodies, followed by horseradish peroxidase-conjugated streptavidin (ZSGB-BIO, Beijing, China). Image J Pro-Plus 6.0 (Media Cybernetics, MD, USA) was used to analyze the histopathological features. The positive staining areas were expressed as percentages of the stained areas divided by the total intima areas in at least 15 high amplification fields (200×).

For immunofluorescence, the cells and tissue sections were first stained with antibodies (1:200) specific for NF90/NF110 (Abcam, ab92355), NF90 (Santa Cruz, sc-377406), AGEs (Abcam, ab23722), FBXW7 (Abcam, ab109617), AGER1 (Proteintech, 14916-1-AP; Santa, sc-74408), OPN (Abcam, ab8448; Santa, sc-21742), α-SMA (Sigma, A5228), and Runx2 (Abcam, 192256), and then stained with fluorescently labeled secondary antibodies (ZSGB-BIO, Beijing, China). Nuclei were counterstained with 4,6-diamidino-2-phenylindole (DAPI; Invitrogen, Carlsbad, CA, USA). The images were analyzed using LSM 710 laser confocal microscope (Carl Zeiss AG, Oberkochen, Germany) equipped with ZEN 2009 Light Edition software (Carl Zeiss AG, Oberkochen, Germany).

### Plasmid, siRNA, and virus
The expression vectors of Flag-tagged full-length homo NF90 and NF110 and Myc-tagged homo AGER1 were generated by subcloning their full-length cDNAs into Flag-CMV$_{10}$ vector and Myc-CMV$_{10}$ vector, respectively, as previously described[56]. The ubiquitin gene was subcloned into pCGN-HA vector. The siRNA sequences for FBXW7, NF90, and NF110 were as follows: si-FBXW7: 5′-GACGCCGAAUUACAUCU-GUTT-3′, si-NF90: 5′-CAGCGUUGUUCGGCAUCAA-3′, si-NF110: 5′-GGA

UGUUGUCACAGCUAGU-3′. Lentiviruses for NF90/110 knockdown (NF90/110 shRNA), lentiviruses for NF90/110 overexpression, and adenovirus expressing FBXW7 (Ad-FBXW7) were generated by Genechem Co., Ltd (Shanghai, China).

## Cell culture, treatment, and transfection
HAVSMCs (CRL-1999) and HEK293T cells (CRL-11268) were obtained from American Type Culture Collection (ATCC, Rockefeller, MD, USA). HAVSMCs were cultured in smooth muscle cell media (SMCM, 5.5 mM glucose; Sciencell, San Diego, CA, USA) supplemented with 1% smooth muscle cell growth supplement, 2% fetal bovine serum (FBS), and 1% penicillin/streptomycin solution. HEK293T cells were cultured in Dulbecco's modified Eagle's medium (DMEM, 5.5 mM glucose; Gibco, Grand Island, NY, USA) supplemented with 10% FBS and 1% penicillin/streptomycin solution. Aortic primary VSMCs and peritoneal macrophages were isolated from mice as described previously[57]. Aortic primary endothelial cells were isolated from mice as described previously[58]. All cells were incubated at 37 °C in an incubator with 5% $CO_2$. The osteogenic medium with 10 mM β-glycerophosphate (β-GP, Sigma-Aldrich, St. Louis, MO, USA) was used to induce VSMCs calcification for one to twenty-one days. The cultured HAVSMCs were treated with normal glucose (NG; 5.5 mM) or high-dose glucose (HG; 27.5 mM; Sigma-Aldric, St. Louis, MO, USA), BSA (200 μg/ml; Biovision, Waltham, MA, USA) or AGE-BSA (200 μg/ml; Biovision, Waltham, MA, USA) for one to five days to mimic DM.

Cycloheximide (CHX) (50 μg/ml; Selleck, Shanghai, China) and MG132 (10 μM; MCE, Merced, CA, USA) were used to inhibit protein translation and proteasome pathway for indicated times, respectively. Actinomycin D (AcmD, 5 μg/ml; APExBIO, Houston, TX, USA) was used to block transcription for indicated times in HAVSMCs. Chloroquine (CQ, 10 μM, MCE, Merced, CA, USA) and 3-Methyladenine (3-MA, 10 mM, Selleck, Shanghai, China) were used for 48 h to inhibit lysosomal degradation and autophagy. A neutralizing anti-RAGE antibody (Abcam, Cat# ab89911, 20 μg/ml) was used for 24 h to block AGEs-RAGE signal pathway.

Plasmids (1–2 μg/well (6-well plate)) and siRNA duplex (150 pmol/well (6-well plate)) were transfected using Lipo3000 and P3000 (Invitrogen, Carlsbad, CA, USA) in Opti-medium (Gibco, Grand Island, NY, USA) to HAVSMC or HEK293T cells. After 6 h, the medium was exchanged with fresh medium. mRNA levels drop after 24 h and protein levels drop after 48 h. Lentiviruses and adenoviruses were incubated with HAVSMCs to overexpress NF90/110 and FBXW7 or knockdown NF90/110 at a multiplicity of infection (MOI) of 10 for 24 h. mRNA levels drop after 48 h and protein levels drop after 72 h.

## Determination of calcification
HAVSMCs were fixed in 70% ethanol after treatment with 10 mM β-GP for twenty-one days after different treatments. Then, cells were rinsed twice with phosphate buffered saline (PBS, Gibco, Grand Island, NY, USA) and stained with 2% Alizarin-Red solution (Sigma-Aldrich, St. Louis, MO, USA) for 5 min at 20 °C. Aortic roots were fixed in neutral buffered formalin for 90 min, then the tissues were embedded upright in optimal cutting temperature compound (OCT), and frozen sections (4 μm) were stained for calcification with the Alizarin-Red method and counterstained with hematoxylin and eosin. The percentage calcified area was measured using Axiovision image analysis software (Release 4.5), in which two color separation thresholds were used to measure the total tissue area.

## Proteomic sequencing
HAVSMCs were transfected with NF90/110 shRNA then cultured with HG for four days (HG+NC shRNA group and HG+NF90-shRNA group, n = 3 per group). Cells were collected in PBS, and proteins were extracted using lysis buffer (50 mM $NH_4HCO_3$, 2% sodiumdeoxycholate, 25 mM NaCl, pH 8.5). After protein digestion and desalination, each sample's eluents were combined and lyophilized. The lyophilized powder was dissolved in 10 μl solution A (100% distilled water and 0.1% methanoic acid) and centrifuged. The supernatant (1 μg) was analyzed using liquid chromatography-tandem mass spectrometry (LC-MS/MS, Orbitrap Exploris 480). The elution conditions for LC-MS are shown: solution B (100% acetonitrile and 0.1% methanoic acid) 28% for 53 min; 35% for 61 min; 100% for 64 min and 100% for 70 min. Ion source: Nanospray Flex™ (NSI); Ion spray voltage: 2.1 kV; Ion transport tube temperature: 320 °C; Data-dependent acquisition mode; Full scanning range of MS: 350–1500 m/z; Primary resolution: 60,000 (200 m/z); C-trap maximum capacity: custom; C-trap maximum injection time: 50 ms; Second resolution 15,000 (200 m/z); C-trap maximum capacity: custom; C-trap maximum injection time: 22 ms. Peptide fragmentation collision energy is 28% and threshold intensity is $5.0 \times 10^4$ to generate raw data for MS detection (.raw). The resulting spectra were searched by proteome Discoverer 2.4 software (ThermoFisher, Waltham, MA, USA) and Uni-Prot Human, then the 74,854 sequences of interest were searched in the database. KEGG enrichment analyses of differential expression proteins were performed by Metascape. Proteomic sequencing was performed by BioMiao Biological Technology Co., Ltd (Beijing, China).

## RNA immunoprecipitation sequencing
HAVSMCs were incubated with BSA or AGE-BSA for 24 h and fixed using 1% methanal for 10 min. The reaction was stopped using 2.66 M glycine, and then cells were collected and lysed using RIPA lysis buffer. After adding A/G agarose beads and NF90 antibody (Santa cruz, sc-377406) to the lysis buffer, the mixture was incubated for 2 h at 20 °C. The protein A/G agarose beads were washed several times and digested using 90% proteinase K. Subsequently, the pulled-down RNA was purified. RNA-seq strand-specific libraries were constructed using the TruSeq® Stranded Total RNA Sample Preparation kit (Illumina, San Diego, CA, USA). Qubit® 2.0 Fluorometer (Life Technologies, Carlsbad, CA, USA) was used to quantify purified libraries, and Agilent 2100 bioanalyzer (Agilent Technologies, Palo Alto, CA, USA) was used to confirm the insert size. A cluster was generated by cBot and sequenced on the HiSeq X Ten (Illumina, San Diego, CA, USA). The raw reads of RNA immunoprecipitation-seq were analyzed by high-quality mapped reads (MAPQ ≥ 30). The library construction and sequencing were provided at Sinomics Corporation (Shanghai, China).

## ALP activity and calcium content assay
ALP activity was detected using ALP Assay Kit (Jiancheng Bioengineering Institute, Nanjing, China). The cells were collected with PBS and then were broken by ultrasound. The suspension were measured according to the instructions and data was acquired using a microplate reader with wavelength 520 nm (Bio-Rad, Hercules, CA, USA). Calcium content was detected using Calcium Assay kit (Jiancheng Bioengineering Institute, Nanjing, China). After the cells were collected with deionized water, the cells were broken by ultrasound. The suspension were measured according to the instructions and data was acquired using a microplate reader with wavelength 610 nm (Bio-Rad, Hercules, CA, USA). Standard curves were plotted to calculate concentrations of the calcium and ALP in samples.

## Quantification of MV
VSMC vesicles were extracted and determined from the medium as previously described[59]. The VSMC medium was harvested and digested by collagenase for 5 min at 37 °C, then the medium was centrifuged at $10,000 \times g$ for 30 min at 4 °C to remove cells and apoptotic bodies. The MVs were then collected from the supernatant by centrifugation at $100,000 \times g$ for 30 min at 4 °C and resuspended with 1% Triton X-100. BCA protein assay kit (Solarbio, Beijing, China) was used to determine protein content. An increase in protein level was indicative of an increased number of MVs.

## Western blotting

Proteins extracted from HAVSMCs were lysed using RIPA lysis buffer (Solarbio, Beijing, China). Lysed proteins were loaded and separated by 10% SDS-PAGE and transferred to PVDF membranes. After blocking in 5% non-fat milk for 1 h, the membranes were incubated with primary antibodies (1:1000) against NF90/110 (Abcam, ab92355), NF90 (Santa cruz, sc-377406), β-actin (Sigma, SAB2100037), Msx2 (Abcam, ab227720), Runx2 (Cell Signalling Technology, 12556), BMP2 (Novus, NBP1-19751), α-SMA (Sigma, A5228), OPN (Abcam, ab8448), calponin (Abcam, ab46794), Bax (Cell Signaling Technology, 2772S), Bcl-2 (Affinity biosciences, AF6139), Cleaved-Caspase-3 (Abcam, ab2302), AGER1 (Proteintech, 14916-1-AP), ubiquitin (Proteintech, 10201-2-AP), HA-tag (Cell Signaling Technology, 3724), Myc-Tag (Cell Signaling Technology, 2278), Flag-Tag (Sigma, F2555), p-STAT1 (Cell Signaling Technology, Cat# 9167), STAT1 (Cell Signaling Technology, Cat# 14994), p-smad1/5 (Cell Signaling Technology, Cat# 9516), Smad1/5 (Abcam, ab300164), p-NF-κB p65 (Cell Signaling Technology, Cat# 3033S), NF-κB p65 (Cell Signaling Technology, Cat# 6956S), p-p38 (Cell Signaling Technology, Cat# 4511S), p38 (Cell Signaling Technology, Cat# 8690S), p-ERK1/2 (Cell Signaling Technology, Cat# 4370S), ERK1/2 (Cell Signaling Technology, Cat# 4695S), p-AKT (Cell Signaling Technology, Cat# 4060S), AKT (Cell Signaling Technology, Cat# 4691S), FBXW7 (Abcam, ab109617), HECTD1 (Abcam, ab101992), ARIH1 (ABclonal, A17123) and HUWE1 (Abcam, ab271032) and horseradish peroxidase-conjugated goat anti-mouse or anti-rabbit secondary antibodies (ZSGB-BIO, Beijing, China). Proteins were visualized using Amersham Imager 680.

## RT-qPCR

Total RNA from HAVSMC was prepared using an RNA extraction kit (Takara, saka-shi, Japan), and 1 μg RNA was reverse-transcribed into cDNA using HiScriptIIIRT SuperMix kit (Vazyme, Nanjing, China). Then SYBR Green Master mix kit (Roche, Basel, Switzerland) was used to carry out Real-Time PCR on a Real-Time PCR System (LightCycler 96, Roche, Basel, Switzerland). All conditions and reagents of qPCR follow the kit manufacturer's instructions. The average cycle thresholds (Ct) were used to determine the mRNA expression. The relative change of mRNA was calculated using the 2-ΔΔCT method. The primer sequences are listed in (Supplementary Table 4).

## TUNEL staining

In Situ Cell Death Detection Kit (Roche, Basel, Switzerland) was administered to test VSMC apoptosis in paraffin-embedded sections and cultured VSMCs according to the manufacturer's specifications. The fluorescent pictures were obtained with a fluorescence microscope.

## Flow cytometric analysis

After exposure to different intervenes, HAVSMCs were double stained with PE Annexin V Apoptosis Detection Kits I (BD Biosciences, San Jose, CA, USA). BD FACS Calibur (BD Biosciences, San Jose, CA, USA) was used to examine the fluorescence intensity.

## Co-IP

Protein-protein Co-IP was performed using an IP Assay kit (Proteintech, Wuhan, China) following the manufacturer's instructions. Protein-protein complexes were extracted from HAVSMCs or HEK293T cells. For immunoprecipitation, antibodies (1 μg/mg protein) against NF90 (Santa Cruz, sc-377406), ubiquitin (Proteintech, 10201-2-AP), OST48 (AGER1; Santa Cruz, sc-74408), FBXW7 (Abcam, ab109617), Myc- Tag (Cell Signaling Technology, 2278), and normal mouse or rabbit IgG (1 μg/mg protein) were used.

## RNA immunoprecipitation

RNA-Binding Protein Immunoprecipitation kit (Millipore, Boston, MA, USA) was used to perform RNA immunoprecipitation assay following manufacturer's instructions. NF90/110-binding RNA complexes with anti-NF90 antibody (Santa Cruz, sc-377406, 5 μg/2.0ˆ10⁷ cells) were extracted from HAVSMCs and tested by PCR. Specific primers targeting the 3'-untranslated region (3'-UTR) of FBXW7 mRNA are listed in Supplementary Table 4.

## FBXW7 mRNA stability analysis

HAVSMCs were treated with actinomycin D (5 μg/ml) for 0, 1, 2, 3, 4, and 5 h. Total RNA was isolated from cells using an RNA extraction kit (Takara, saka-shi, Japan) according to the manufacturer's instructions. The total RNA was reverse-transcribed using the HiScriptIIIRT Super-Mix kit (Vazyme, Nanjing, China), and qRT-PCR was performed using the LightCycler 480 SYBR Green I Master (Roche, Basel, Switzerland). The FBXW7 mRNA levels were evaluated by reverse-transcription cDNA and RT-qPCR. The primer sequences targeting FBXW7 mRNA are listed in Supplementary Table 4.

## Statistical analysis

SPSS statistical software (SPSS Inc., Chicago, IL) and GraphPad Prism 8 (GraphPad Software, San Diego, CA, USA) were used for statistical analyses. Data are presented as the mean ± SEM. The statistical differences were evaluated using Student's $t$-test for data classified into two groups and one-way ANOVA was used for data classified into three or more groups (Tukey's post-test). The assumption of normality was tested by the Shapiro-Wilks test. All the replicate experiments are biological replicates with at least three times and all statistical tests were two-tailed. $p < 0.05$ was considered statistically significant.

## Reporting summary

Further information on research design is available in the Nature Portfolio Reporting Summary linked to this article.

## Data availability

The RNA immunoprecipitation-seq data generated in this study have been deposited in the NCBI Gene Expression Omnibus (GEO) under accession code GSE253682. The proteome-seq data generated in this study have been deposited in iProX under accession code PXD048707. All other data generated or analyzed in this study are included in this published paper and its Supplementary Information file/Source Data file. Additional data related to this paper are available from the corresponding author on request. Source data are provided with this paper.

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

## Acknowledgements

This work was supported by the National Natural Science Foundation of China (#82370455 to M.Z., #82302429 to F.X.), the National Key R & D program of China (#2017YFC0908700 to M.Z., 2017YFC0908703 to M.Z.), the Taishan Scholar Project of Shandong Province of China (No. ts20190972 to M.Z.), the Postdoctoral Science Foundation of China (No.2022M721964 to F.X.) and the Clinical Research Project of Shandong University (2020SDUCRCA016 to M.Z.).

## Author contributions

F.X., B.L., W.Q., J.H., B.X., Y.C., and M.Z. designed and performed the research, Z.W., Y.H., J.C., X.Z., and M.Z. analyzed data, F.X., W.Q., Z.W., Y.Z., Y.C., Y.Z., and M.Z. conceived the project, reviewed the data and wrote the manuscript.

## Competing interests

The authors declare no competing interests.
