## [Peer Review File · Nature Communications]

Smooth muscle NF90 deficiency ameliorates diabetic atherosclerotic calcification in male mice via FBXW7-AGER1-AGEs axisREVIEWER COMMENTS

Reviewer #1 (Remarks to the Author):

In this manuscript, using both diabetic/hyperlipidemic mouse models and high-glucose-treated SMCs, Xie and colleagues showed that the mechanism of the AGE accumulation in the artery wall is due to AGE-induced degradation of AGER1 via the NF90/FBXW7 pathway. The work is novel, the experimental design is thorough, and the manuscript is well written. The findings have potential to add significance to the field. However, the rescue of AGER1 expression inhibited by DM in NFSM-KO requires more convincing data. The following comments need to be addressed.

1. The rescues of AGER1 (Fig. 2E, 7G) and FBXW7 (Fig. 6G) in the NFSM-KO DM mice are not convincing. The expressions in the subendothelial layer are still increased in these mice.
2. In Abstract, please revise sentence (Lines 34 – 40) by removing "..., thereby inhibiting endocytosis and degradation of AGEs ..." since it lacks the supporting data.
3. In addition, line 34 – "Mechanically ..." should be replaced with "Mechanistically ..."
4. Please add the discuss on how AGEs are cleared physiologically after binding to AGER1. This would provide clarification as to how AGEs get accumulated when AGER1 are degraded intracellularly through ubiquitination induced by FBXW7. In the case of AGER1 degradation induced by diabetes, are there insights into how AGEs cleared from plasma?
5. In human specimen, NF90/110 appears to locate in the medial layer whereas in the mice, the staining shows subendothelial layer. Please add this difference to the results section and, if possible, add discussion.
6. Please add to the discussion the order of signal transduction suggested by the findings in Fig 7G. While AAV-FBXW7 further decreased AGER1 expression in the control (Nff/f), in NFSM-KO mice, overexpression of FBXW7 did not rescue the DM-induced AGER1 expression, calcification, and other markers, suggesting that FBXW7 is located upstream of NF90/110.
7. In Figures,
 1. Please provide higher magnification images as inserts in Figure 1A. The human arteries do not look to have atherosclerosis or calcification. Is there any information on the DM status in these patients and the location of calcification with respect to the images shown in Fig. 1A?
 2. Please indicate in the figure and/or in the legend that NF90/110 shRNA is a knockdown.
 3. "Calcium concentration" "calcium levels (e.g. line 170)" in the figures or text should be revised to "matrix calcification or matrix calcium content."
 4. Please clarify what "calcified ratio" is.
 5. NF90/110 should also be included in Fig. 6J, in addition to ILF3.
 6. The cell on the left side of the Fig. 8 should be enlarged to clearly depict the bulk of the data in this manuscript. The label "AGEs Endocytosis & Degradation" should be removed since it would not lead to AGE accumulation intracellularly.

Minor comments

1. Please state the concentration of glucose in the DMEM used as well as the total concentration of glucose after 25 mM was added to treat the cells.
2. Please provide source and content of the high-fat diet (line 526).
3. Figure numbers should be included on the Figures.
4. Please define "RIP".
5. Please replace "osteocyte" with "osteoblastic" in line 353 since HAVSMC did not look like osteocytes.

Reviewer #2 (Remarks to the Author):

In this study, the authors investigated the underlying mechanisms which hyperglycemia accelerates calcification of atherosclerotic plaques in diabetic patients using vascular smooth muscle cells (VSMCs), human samples and mouse models with a variety of genetic modifications. This study focused on the role of vascular smooth muscle cell nuclear factor 90/110 (NF90/110) in mediating hyperglycemia-induced AGE accumulation and leading to enhancement of diabetic atherosclerotic calcification. Their results demonstrated that hyperglycemia dramatically increased

VSMC NF90/110 activation both in human and mouse atherosclerotic calcified tissues with diabetes, which modulated FBXW7/AGER1/AGEs pathway, resulting in the prolonged accumulation of AGEs and VSMC calcification. The authors claimed that both hyperglycemia and AGEs accelerated diabetic atherosclerotic calcification by inducing VSMC phenotypic changes to osteoblast-like cells, VSMC apoptosis, and matrix vesicle release. In general, the study designs and the methodologies were sound. The interpretations of results were also reasonable. This study provided novel knowledge for this field but it is not a flawless study. Before it can be considered to be published in this journal, the following concerns need to be addressed:

1. Frankly, this study mainly focused on the underlying mechanism which hyperglycemia stimulates AGE accumulation but not the mechanism which HG/AGEs stimulate SMC phenotypic switch from contractile to osteoblastic, leading to calcification in atherosclerotic lesions. For example, the only data they had were increased osteoblastic marker, OPN and Vimentin, etc. By the way, Vimentin is not a good marker for osteoblasts. This part of study lacks mechanistic data;
2. Although AGE may mediate certain effects of high glucose, hyperglycemia definitely plays more roles than AGE in mediating the manifestations of DM. Therefore, the statement related them needs to be clarified clearly;
3. High glucose induced the accumulation of AGEs, which leads to SMC phenotypic switch. However, this is not the only reason for enhanced atherosclerotic calcification. This needs to be addressed in the discussion section. In addition, based on the results, AGEs are able to enhance AGER1 degradation, leading to more accumulation of AGEs, which makes the worse scenario. This is feedback effect, which can be considered as the second wave of hitting. Therefore, they need to be presented separately or organized well. For example, focusing on HG functions and underlying mechanism first, such as whether HG-enhanced AGER1 degradation is also dependent on FBXW7 stability. The effect of AGEs on this pathway is the second stage studies. This also needs to be addressed in the discussion section;
4. Figure 1G, Runx2 densitometry data in HG treated NF90 ShRNA group (the 4th lanes) did not reflect WB results;
5. Figure 1H, the Alizarin staining did not show significantly difference between Non-DM and DM in NF flox/flox mice but the densitometry data showed the significantly enhanced in DM group;
6. Figure 2F, Runx2 densitometry data was inconsistent with WB results (lane1 vs lane 3 and 4);
7. Figure 3A, α -SMA and calponin densitometry data were inconsistent with WB results (lane 3 vs 4);
8. Figure 3B α -SMA staining results looked different from supplementary figure 2 in NF flox/flox mice, particular in DM panel. In addition, why α -SMA signal was so weak in non-DM NF flox/flox mice in supplemental figure 2;
9. FBXW7 input was much higher in the AGE-BSA group (the 3rd lane of supplementary Figure 3B) than other groups;
10. Figure 4J, K, L and Figure 5 H, I, J, K, L need to provide protein ladder information;
11. Figure 5C, AGER1 densitometry data was inconsistent with WB results;
12. Figure 5K needs NF90 input information and FBXW7 input data (the 3rd lane) looked weird;
13. Figure 7G, the AGER1 staining results (NF-SM KO animal vs NF-flox/flox animal) did not support the densitometry data;
14. Supplementary Figure 4 presentation and results looked weird/confusion. First, OPN and Bax2 results were listed 3 times and FBXW7, AGER1 and Runx2 results were listed 2 times. Second, almost all checked markers staining results looked difference between NF SM-KO Non-DM and NF flox/flox Non-DM. Some of them were inconsistent between panels, such as Runx2 staining results

between panel C and F. Third, the differences between Non-DM and DM in NF flox/flox group were minimal or invisible in several markers, such as Bax2 in panel D, E and F; FBXW7 in panel D; Runx2 in panel F.

15. Line 212 or 219 or 245, instead of using "a reversible effect" or "inhibited" or "cancelled", using "preventing or abolishing the effect of AGEs" will be better;

16. Line 662, please make sure the slides were 4 mm section;

17. For mice studies, in the methods part, the authors claimed 12 mice for each group but, in the results section, N=10. In addition, STZ-induced diabetes mice normally are not easy to maintain for more than 6 weeks, why did these mice need to be maintained for 12 weeks? And how?

Itemized Responses to Reviewers Comments

Reviewer #1 (Remarks to the Author):

Major concerns:

Q1. The rescues of AGER1 (Fig. 2E, 7G) and FBXW7 (Fig. 6G) in the NFSM-KO DM mice are not convincing. The expressions in the subendothelial layer are still increased in these mice.

A1. We appreciate the careful observation that the reviewer has made. We repeated these experiments and demonstrated the expressions of AGER1 and FBXW7 in the plaque of NF^{SM-KO} DM mice. The new better images have presented in Figure 2e, 7g and Figure 6g.

Q2. In Abstract, please revise sentence (Lines 34 – 40) by removing “..., thereby inhibiting endocytosis and degradation of AGEs ...” since it lacks the supporting data.

A2. Thanks for your suggestion. We have removed the sentence from the abstract (page 2, line 36).

Q3. In addition, line 34 – “Mechanically ...” should be replaced with “Mechanistically ...”

A3. Thank you for the catch. We have replaced “Mechanically” with “Mechanistically ...” (page 2, line 34).

Q4. Please add the discuss on how AGEs are cleared physiologically after binding to AGER1. This would provide clarification as to how AGEs get accumulated when AGER1 are degraded intracellularly through ubiquitination induced by FBXW7. In the case of AGER1 degradation induced by diabetes, are there insights into how AGEs cleared from plasma?

A4. We appreciate the suggestion by the reviewer. We have added “how AGEs are cleared physiologically after binding to AGER1” in the discussion section (page 22, line 471-477). For the question raised by the reviewer “In the case of AGER1 degradation induced by diabetes, are there insights into how AGEs cleared from plasma?”. Upon further research, we noted that at present, there is little information about the molecular processes and factors involved in AGEs cleared from plasma under diabetes. During normal physiological conditions, mononuclear macrophages

can take up plasma AGEs and degrade them into AGEs polypeptides, which may be carried out by endocytosis after non-specific binding [1, 2]. Mononuclear cells can also activate the extracellular protein lysin system by secreting cytokines. Therefore, AGEs in plasma usually exist in the form of AGEs polypeptides, and the AGEs polypeptides formed by degradation normally rely on kidney clearance [1, 2]. Moreover, macrophage scavenger receptor group A, SR-A, also appears to function as a negative regulator of the level of AGEs in plasma [3]. In addition, circulating monocyte AGER1 levels correlate strongly with systemic AGEs, independent of age in healthy adults [4]. Vlassara et al. have also shown that the AGE-R1/OST-48 level in peripheral blood mononuclear cell (PBMCs) positively correlates with plasma and urine AGEs as well as with oxidative stress markers in healthy participants and negatively in patients with stage 3 chronic kidney disease (CKD-3) [5]. These data support the point that the level of receptor AGER1 in circulating monocyte is correlate strongly with AGEs clearance from plasma. Despite multiple advances in AGEs cleared from plasma during normal physiological conditions, many gaps remain in AGEs metabolism-related diabetic pathophysiology that need to be further elucidated.

Q5. In human specimen, NF90/110 appears to locate in the medial layer whereas in the mice, the staining shows subendothelial layer. Please add this difference to the results section and, if possible, add discussion.

A5. *We appreciate the careful observation that the reviewer has made. We think the reason for this difference of NF90/110 location between human and mice is due to the mouse medial layer of aortic root being too thin and not obvious. According to the reviewer suggestion, we re-described the results of Figure. 1a and b, and explained the reason in results. (page 5, line 99-106).*

Q6. Please add to the discussion the order of signal transduction suggested by the findings in Fig 7G. While AAV-FBXW7 further decreased AGER1 expression in the control (Nff/f), in NFSM-KO mice, overexpression of FBXW7 did not rescue the DM-induced AGER1 expression, calcification, and other markers, suggesting that FBXW7 is located upstream of NF90/110.

A6. We appreciate the question by the reviewer. We added the order of signal transduction suggested by the findings in Fig 7g into the discussion section (page 21, line 439-450). We repeated the experiment of AAV-FBXW7 overexpression in NF^{SM-KO} mice, and the IHC staining showed that in NF^{SM-KO} mice (lane 4 vs 2), overexpression of FBXW7 rescued the DM-induced AGER1 expression, calcification, and other markers, suggesting that FBXW7 is located downstream of NF90/110. They are now presented in Fig7g.

Q7. In Figures,

Q1. Please provide higher magnification images as inserts in Figure 1A. The human arteries do not look to have atherosclerosis or calcification. Is there any information on the DM status in these patients and the location of calcification with respect to the images shown in Fig. 1A?

A1. Thank you very much for your kind reminder. The higher resolution and magnification images have been shown in Figure 1a. Alizarin-red staining results of human coronary arteries were showed in Figure 1a and reflected the calcium deposition. The information of the patients with or without DM was listed in Table.3, including BG, HbA1c, BW, FINS, TG, TC, LDL, HDL, calcium and phosphorus.

Q2. Please indicate in the figure and/or in the legend that NF90/110 shRNA is a knockdown.

A2. We appreciate the careful observation that the reviewer has made and we have revised the legend (page 42, line 1010; page 43, line 1013 and 1032; page 44, line 1045-1046; page 45, line 1056 and 1062).

Q3. “Calcium concentration” “calcium levels (e.g. line 170)” in the figures or text should be revised to “matrix calcification or matrix calcium content.”

A3. Thank you for the suggested edit. We have now gone through the figures and text and revised for clarity (page 6, line 117; page 8, line 168; page 43, line 1015; page 44, line 1039-1040, 1049).

Q4. Please clarify what “calcified ratio” is.

A4. Thank you for the request. Here the “calcified ratio” means the relative Alizarin red-positive area. In order to avoid confusion for readers, we have remodified “calcified ratio” in Figure 1e and 2m to relative Alizarin red-positive area.

Q5. NF90/110 should also be included in Fig. 6J, in addition to ILF3.

A5. As suggested by the reviewer, we have NF90/110 completed in Fig.6j.

Q6. The cell on the left side of the Fig. 8 should be enlarged to clearly depict the bulk of the data in this manuscript. The label “AGEs Endocytosis & Degradation” should be removed since it would not lead to AGE accumulation intracellularly.

A6. This is an excellent point. We have revised the Fig.8 according the reviewer’s suggestion.

Minor comments

Q1. Please state the concentration of glucose in the DMEM used as well as the total concentration of glucose after 25 mM was added to treat the cells.

A1. We have now added the concentrations of glucose in the SMCM and DMEM as well as total concentration of glucose (page 29, line 626, 629; page 30, line 636-637).

Q2. Please provide source and content of the high-fat diet (line 526).

A2. We have done that according to the reviewer’s suggestions (page 26, line 561-563).

Q3. Figure numbers should be included on the Figures.

A3. Thank you very much for your kind reminder. We have done that.

Q4. Please define "RIP".

A4. We have done that (page 14, line 293).

Q5. Please replace “osteocyte” with “osteoblastic” in line 353 since HAVSMC did not look like osteocytes.

A5. Thank you very much for your kind reminder. We have replaced “osteocyte” with “osteoblastic” (in page 19, line 395; in page 20, line 419).

Reviewer #2 (Remarks to the Author):

Q1. Frankly, this study mainly focused on the underlying mechanism which hyperglycemia stimulates AGE accumulation but not the mechanism which HG/AGEs

stimulate SMC phenotypic switch from contractile to osteoblastic, leading to calcification in atherosclerotic lesions. For example, the only data they had were increased osteoblastic marker, OPN and Vimentin, etc. By the way, Vimentin is not a good marker for osteoblasts. This part of study lacks mechanistic data;

A1. Indeed, our experimental data in the current study mainly focused on the underlying mechanism which NF90/110 stimulates AGEs accumulation. For the mechanism which HG/AGEs stimulate SMC phenotypic switch from contractile to osteoblastic, leading to calcification in atherosclerotic lesions, there have been hundreds of studies showing that the engagement of AGEs with its chief cellular receptor, RAGE, activates a myriad of signaling pathways such as MAPK/ERK1/2, TGF- β , JAK, and NF- κ B, leading to VSMCs phenotypic switch and calcium deposition [6, 7]. As the suggested by reviewer, we carried out additional experiments and analyzed the effect of NF90/110 on AGEs-mediated downstream signaling pathways participating in atherosclerotic calcification (page11-12, line 233-252). The results are presented in the supplemental Fig. 4a and b. Our results showed that VSMC NF90/110 deletion inhibited AGEs-mediated downstream signaling pathways. overexpression of NF90/110 and AGE treatment much higher raised the levels of p-p38, p-ERK1/2, p-AKT, p-Smad1/5, p-NF- κ B, and p-STAT1 compared to those treated with AGEs alone. Further, we observed that the interaction of AGE and RAGE that was inhibited by the anti-RAGE antibody obviously reversed the effect of overexpression of NF90/110. Based on our findings, we could speculate that VSMC NF90/110 mediated AGE accumulation to accelerate vascular calcification by activating multiple signaling pathways. In relation to the second question, we re-selected BMP2 as the new osteoblastic marker in present study (Figure. 3a and 7a).

Q2. Although AGE may mediate certain effects of high glucose, hyperglycemia definitely plays more roles than AGE in mediating the manifestations of DM. Therefore, the statement related them needs to be clarified clearly;

A2. We appreciate the question by the reviewer, and indeed diabetes is a group of disorders of metabolic abnormalities, whose main feature is chronic hyperglycemia

that results to vascular complications [8]. Hyperglycemia is closely connected with atherosclerosis through several pathological pathways, such as oxidative stress, AGEs generation, protein kinase C (PKC) signaling, chronic inflammation, circulating non-coding RNAs, and epigenetic modification [9]. Calcified lesions in atherosclerotic plaques are a known predictive factor of future cardiovascular events and a marker of poor prognosis and resistance to coronary intervention therapies [10]. Hyperglycemia accelerates the calcification of atherosclerotic plaques in diabetic patients [11, 12]. The metabolic imbalance of AGE plays a central role in the pathophysiological processes that lead to the development of diabetic atherosclerotic calcifications [13]. In this study, we demonstrated for the first time that AGEs-mediated NF90 activation causes ubiquitination and proteasome degradation of AGER1 by enhancing the mRNA stability of FBXW7, which in turn leads to the accumulation of AGEs in VSMCs and accelerates atherosclerotic calcification. As the suggested by the reviewer, we have clarified on this question in our discussion (page 20-21, line 429-442).

Q3. High glucose induced the accumulation of AGEs, which leads to SMC phenotypic switch. However, this is not the only reason for enhanced atherosclerotic calcification. This needs to be addressed in the discussion section. In addition, based on the results, AGEs are able to enhance AGER1 degradation, leading to more accumulation of AGEs, which makes the worse scenario. This is feedback effect, which can be considered as the second wave of hitting. Therefore, they need to be presented separately or organized well. For example, focusing on HG functions and underlying mechanism first, such as whether HG-enhanced AGER1 degradation is also dependent on FBWX7 stability. The effect of AGEs on this pathway is the second stage studies. This also needs to be addressed in the discussion section.

A3. This is an excellent point. As the suggested by the reviewer, we have added explanations in the discussion in relation to “the reasons of HG enhancing atherosclerotic calcification apart from affecting AGEs” (page20-21, line 434-439).

For “This is feedback effect”, we gave a clear description in discussion section (page 21, line 439-450). Also, we appreciate the reviewer's point about a "second wave of

strikes". For this question, we have now carried additional experiments to demonstrate that "whether HG-enhanced AGER1 degradation is also dependent on FBWX7 stability" (Supplementary Fig. 3 and 6). The results showed that HG and AGEs significantly improved the levels of NF90/110 and FBWX7 in HAVSMCs and promoted the degradation of AGER1, but the change time nodes of NF90/110, FBWX7 and AGER1 were different under HG and AGEs treatment. NF90/110 and FBWX7 elevations and AGER1 degradation were on Day Four and Five after HG stimulation (Supplementary Fig. 3a and 6a). Moreover, we found that the levels of AGEs in cell culture-medium were also significantly elevated on Day Four and Five after HG stimulation (Supplementary Fig. 3b). Further, we observed that NF90/110 and FBWX7 levels increased, and AGER1 was decreased significantly on 24 hours after AGE stimulation, much earlier than HG stimulation (supplementary Figure 3c and 6b). In sum, these results indicated 1-3 days HG stimulation had no obvious effect on NF90/110-FBXW7-AGER1 axis before AGEs generation, and the effect of HG on NF90/110-FBXW7-AGER1 axis is achieved by inducing the production of AGEs. So, AGEs not HG-enhanced AGER1 degradation is dependent on FBWX7 stability (page 9, line 181-191; page 16-17, line 348-353).

Q4. Figure 1G, Runx2 densitometry data in HG treated NF90 ShRNA group (the 4th lanes) did not reflect WB results;

A4. We appreciate the careful observation that the reviewer has made and have re-analyzed the Runx2 densitometry in Figure 1g.

Q5. Figure 1H, the Alizarin staining did not show significantly difference between Non-DM and DM in NF flox/flox mice but the densitometry data showed the significantly enhanced in DM group;

A5. As suggested by the reviewer, we repeated these experiments and demonstrated more striking change of Alizarin staining between Non-DM and DM in NF^{flox/flox} mice (lane 1 vs. 2) . The better image is presented in Figure 1h.

Q6. Figure 2F, Runx2 densitometry data was inconsistent with WB results (lane 1 vs lane 3 and 4);

A6. *We appreciate the careful observation that the reviewer has made. We repeated these experiments and the better images are presented in Figure 2f.*

Q7. Figure 3A, α -SMA and calponin densitometry data were inconsistent with WB results (lane 3 vs 4);

A7. *We appreciate the careful observation that the reviewer has made. We have re-analyzed the α -SMA and calponin densitometry in Figure 3a.*

Q8. Figure 3B α -SMA staining results looked different from supplementary figure 2 in NF flox/flox mice, particular in DM panel. In addition, why α -SMA signal was so weak in non-DM NF flox/flox mice in supplemental figure 2;

A8. *We appreciate the careful observation that the reviewer has made. We repeated the immunofluorescence staining experiment, the better images are presented in supplemental Figure 2.*

Q9. FBXW7 input was much higher in the AGE-BSA group (the 3rd lane of supplementary Figure 3B) than other groups;

A9. *We appreciate the careful observation that the reviewer has made. Due to the addition of some additional experiments, the supplementary Figure 3 has been renamed supplementary Figure 5 in the revised text. In fact, our results showed that AGE treatment markedly increased the expressions of HECTD1, ARIH1, HUWE1 and FBXW7 in HAVSMCs compared with the BSA control (the 3rd lane of supplementary Figure 5b). In addition, we repeated these experiments and better western blot images of HECTD1, ARIH1 and HUWE1 were presented in input of supplementary Figure 5b*

Q10. Figure 4J, K, L and Figure 5 H, I, J, K, L need to provide protein ladder information;

A10. *We appreciate the careful observation that the reviewer has made. We have provided protein ladder in Figure 4j, k, l and Figure 5 h, i, j, k, l.*

Q11. Figure 5C, AGER1 densitometry data was inconsistent with WB results;

A11. *We appreciate the careful observation that the reviewer has made. We re-analyzed the AGER1 densitometry in Figure 5c.*

Q12. Figure 5K needs NF90 input information and FBXW7 input data (the 3rd lane) looked weird;

A12. As suggested by the reviewer, this is now completed and presented in Figure 5k.

Q13. Figure 7G, the AGER1 staining results (NF-SM KO animal vs NF-flox/flox animal) did not support the densitometry data;

A13. We appreciate the careful observation that the reviewer has made. We repeated these experiments and demonstrated the expressions of AGER1 in NF^{SM-KO} DM mice. The new better images have presented in Figure 7g.

Q14. Supplementary Figure 4 presentation and results looked weird/confusion. First, OPN and Bax2 results were listed 3 times and FBXW7, AGER1 and Runx2 results were listed 2 times. Second, almost all checked markers staining results looked difference between NF SM-KO Non-DM and NF flox/flox Non-DM. Some of them were inconsistent between panels, such as Runx2 staining results between panel C and F. Third, the differences between Non-DM and DM in NF flox/flox group were minimal or invisible in several markers, such as Bax2 in panel D, E and F; FBXW7 in panel D; Runx2 in panel F.

A14. This is a valid concern. We removed the Supplementary Figure4 from the revised text.

Q15. Line 212 or 219 or 245, instead of using “a reversible effect” or “inhibited” or “cancelled”, using “preventing or abolishing the effect of AGEs” will be better;

A15. We appreciate the careful observation that the reviewer has made. We have done that (page 11, line 222 and 229; page 13, line 275-276)

Q16. Line 662, please make sure the slides were 4 mm section;

A16. We apologize for the negligence. The slides were 4 μ m section (page 31, line 662)

Q17. For mice studies, in the methods part, the authors claimed 12 mice for each group but, in the results section, N=10. In addition, STZ-induced diabetes mice normally are not easy to maintain for more than 6 weeks, why did these mice need to be maintained for 12 weeks? And how?

A17. Our apologies for the confusion on description of the mice number in the methods part. Twelve mice were selected for each group at the start of the study. However, one or two mice died after in the groups of diabetes. So, we chose n=10

mice for each group at the end of the experiments. To avoid the misunderstanding for readers, we have revised n=10 in the methods part as the suggestion of the reviewer (page). For the question about the STZ-induced diabetes mice model, we did further research. As the reviewer mentioned, high dose (100-200 mg/kg) STZ-induced type 1 diabetes mice are hard to maintain for longer than 6 weeks [14]. In current study, diabetes mice models were induced with low dose of STZ (50 mg/kg), so the diabetes mice were able to be maintained for more than 12 weeks [15, 16]. In addition, the formation of diabetic vascular calcification generally takes 12 weeks in the mice model [17].

Citation

- [1] Mariyam Khalid, Georg Petroianu, Abdu Adem. Advanced Glycation End Products and Diabetes Mellitus: Mechanisms and Perspectives. *Biomolecules*. 2022 Apr 4;12(4):542.
- [2] Chieh-Yu Shen, Cheng-Hsun Lu, Cheng-Han Wu, Ko-Jen Li, Yu-Min Kuo, Song-Chou Hsieh, Chia-Li Yu. The Development of Maillard Reaction, and Advanced Glycation End Product (AGE)-Receptor for AGE (RAGE) Signaling Inhibitors as Novel Therapeutic Strategies for Patients with AGE-Related Diseases. *Molecules*. 2020 Nov 27;25(23):5591.
- [3] Ke Ma, Yiming Xu, Chenchen Wang, Nan Li, Kexue Li, Yan Zhang, Xiaoyu Li, Qing Yang, Hanwen Zhang, Xudong Zhu, Hui Bai, Jingjing Ben, Qingqing Ding, Keran Li, Qin Jiang, Yong Xu, Qi Chen. A cross talk between class A scavenger receptor and receptor for advanced glycation end-products contributes to diabetic retinopathy. *Am J Physiol Endocrinol Metab*. 2014 Dec 15;307(12):E1153-65.
- [4] Z. Yang, Z. Makita, Y. Horii, S. Brunelle, A. Cerami, P. Sehajpal, M. Suthanthiran, H. Vlassara. Two novel rat liver membrane proteins that bind advanced glycosylation endproducts: relationship to macrophage receptor for glucose-modified proteins, *J. Exp. Med*. 174 (1991) 515–524.
- [5] H. Vlassara, W. Cai, S. Goodman, R. Pyzik, A. Yong, X. Chen, L. Zhu, T. Neade, M. Beeri, J.M. Silverman, L. Ferrucci, L. Tansman, G.E. Striker, J. Uribarri,

Protection against loss of innate defenses in adulthood by low advanced glycation end products (AGE) intake: role of the antiinflammatory AGE receptor-1, *J. Clin. Endocrinol. Metab.* 94 (2009) 4483–4491.

[6] Amber M Kay, C LaShan Simpson, James A Stewart Jr. The Role of AGE/RAGE Signaling in Diabetes-Mediated Vascular Calcification. *J Diabetes Res.* 2016; 2016:6809703.

[7] Christiane Ott, Kathleen Jacobs, Elisa Haucke, Anne Navarrete Santos, Tilman Grune, Andreas Simm. Role of advanced glycation end products in cellular signaling. *Redox biology.* 2014; 2:411-429.

[8] American Diabetes Association. Diagnosis and classification of diabetes mellitus. *Diabetes Care.* 2014 Jan;37 Suppl 1:S81-90.

[9] Anastasia Poznyak, Andrey V Grechko, Paolo Poggio, Veronika A Myasoedova, Valentina Alfieri, Alexander N Orekhov. The Diabetes Mellitus-Atherosclerosis Connection: The Role of Lipid and Glucose Metabolism and Chronic Inflammation. *Int J Mol Sci.* 2020 Mar 6;21(5):1835.

[10] Alan Chait, Karin E Bornfeldt. Diabetes and atherosclerosis: is there a role for hyperglycemia? *J Lipid Res.* 2009 Apr;50 Suppl:S335-9.

[11] Kazuyuki Yahagi, Frank D Kolodgie, Christoph Lutter, Hiroyoshi Mori, Maria E Romero, Alope V Finn, Renu Virmani. Pathology of Human Coronary and Carotid Artery Atherosclerosis and Vascular Calcification in Diabetes Mellitus. *Arterioscler Thromb Vasc Biol.* 2017 Feb;37(2):191-204.

[12] Robert H Eckel, Karin E Bornfeldt, Ira J Goldberg. Cardiovascular disease in diabetes, beyond glucose. *Cell Metab.* 2021 Aug 3;33(8):1519-1545.

[13] N C Chilelli, S Burlina, A Lapolla. AGEs, rather than hyperglycemia, are responsible for microvascular complications in diabetes: a "glycoxidation-centric" point of view. *Nutr Metab Cardiovasc Dis.* 2013 Oct;23(10):913-9.

[14] Brian L Furman. Streptozotocin-Induced Diabetic Models in Mice and Rats. *Curr Protoc.* 2021 Apr;1(4):e78.

[15] F Barutta, S Bellini, S Kimura, K Hase, B Corbetta, A Corbelli, F Fiordaliso, S Bruno, L Biancone, A Barreca, M G Papotti, E Hirsh, M Martini, R Gambino, M

Durazzo, H Ohno, G Gruden. Protective effect of the tunneling nanotube-TNFAIP2/M-secl system on podocyte autophagy in diabetic nephropathy Autophagy. 2023 Feb;19(2):505-524.

[16] Sarah J Glastras, Hui Chen, Rachel Teh, Rachel T McGrath, Jason Chen, Carol A Pollock, Muh Geot Wong, Sonia Saad. Mouse Models of Diabetes, Obesity and Related Kidney Disease. PLoS One. 2016 Aug 31;11(8):e0162131.

[17] Peng Li, Ying Wang, Xue Liu, Bin Liu, Zhao-Yang Wang, Fei Xie, Wen Qiao, Er-Shun Liang, Qing-Hua Lu, Ming-Xiang Zhang. Loss of PARP-1 attenuates diabetic arteriosclerotic calcification via Stat1/Runx2 axis. Cell Death Dis. 2020 Jan 10;11(1):22.

REVIEWER COMMENTS

Reviewer #1 (Remarks to the Author):

Xie and colleagues showed that the mechanism of the AGE accumulation and calcification in the artery wall is due to AGE-induced degradation of AGER1 via the NF90/FBXW7 pathway. The work is novel, and the findings have potential to add significance to the field.

In this revised manuscript, the authors have addressed most of the comments in the original review, and the manuscript has improved. However, some key questions remain.

1. Important questions remain to be addressed are as follows.
 - a. The reduction of FBXW7 in the KO aortic section is not convincing. Fig 6G - The quantification for the levels of FBXW7 under DM in fl/fl vs. NF SMKO in the graph still does not match with images.
 - b. Fig 7G - There is a discrepancy in AGER1 expression in the control NF fl/fl mice compared with earlier images (e.g. Fig 2E and 2D).
 - c. To definitively show that the induction of NF90/110 was only due to HG, control cells without HG should be included (or at least at the later time points).
 - d. The authors only used male human specimens and male mice. The title and abstract do not indicate that the findings apply only to one sex. The method section also does not include whether sex and/or gender were considered in the study design.
2. Line 107. The reasoning statement regarding in human medial vs mouse intimal layer is not totally accurate and should be revised. Two different mechanisms govern atherosclerotic vs CKD calcification, neo-intimal vs medial calcification, respectively.
3. Results section is too long (~16 pages), and it can be revised to be more concise.
4. Since the study employed NF90/110 deficiency instead of overexpression, the word "... accelerates ..." should be revised in the title.
5. Please revise the statement in line 519 since this study does not have evidence of increased RAGE expression.
6. Although Figure 8 has improved, the following are needed for clarity.
 1. Please mention in the legend how AGE enters into cells.
 2. Does AGE increase NF90/110 via RAGE signaling?
 3. It would be good to include panel titles in "a" and "b" (e.g. WT, NF90/110 KO, etc.).
 4. In panel a, please remove the arrow from "AGEs" to "More AGEs" and also remove the word "More AGEs." Having the illustration of AGE (depicted as several circles) is self-explanatory.
 5. In the KO cells, please add more AGER-1 on the cell and also remove the word "Lower AGEs."
 6. Please show a bigger difference AGE illustration (e.g 10 circles vs. 3) between the 2 panels.
 7. "Atherosclerotic" should be revised to "matrix calcification."
 8. Panels a and b should depict differences in calcification.

Minor comments

1. Methods for ALP and calcium content assay should be expanded. Please provide brief procedure, substrate name, OD wavelength, and how they were normalized, etc.
2. Line 59 – Since calcium deposition is in the extracellular matrix, the statement should be revised "... stimulated calcium deposition in the extracellular matrix of ... VSMC ..."
3. Fig. 5C. The position of statistical bar in the graph should be properly aligned.
4. Fig. 8. There seem to be typos in "NF100."
5. Line 294 and 761. The word for the acronym RIP should be corrected to "RNA Immunoprecipitation."
6. Line 452... the word "deteriorate" does not fit the statement and should be revised.

Reviewer #2 (Remarks to the Author):

Regarding to authors' rebuttal letter, I was satisfied with most of their answers and noticed the additional experiments they have done. However, for Q14, removing the questionable supplemental Figure 4 without further adjustments is inappropriate.

Regarding to animal gender issue, there was no any information in the title and abstract. It was mentioned that male mice were used.

In addition, L354-L357 has a repeat sentence. Authors should check their MS more carefully.

Itemized Responses to Reviewers Comments

Reviewer #1 (Remarks to the Author):

Q1. Important questions remain to be addressed are as follows.

a. The reduction of FBXW7 in the KO aortic section is not convincing. Fig 6G - The quantification for the levels of FBXW7 under DM in fl/fl vs. NF SMKO in the graph still does not match with images.

A1-a. Thanks for your suggestion. We have re-analyzed the FBXW7 densitometry in Figure 6g.

b. Fig 7G - There is a discrepancy in AGER1 expression in the control NF fl/fl mice compared with earlier images (e.g. Fig 2E and 2D).

A1-b. We appreciate the careful observation that the reviewer has made. We repeated these experiments and the new images have are presented in Figure 7g. The AGER1 densitometry in Figure 7g was re-analyzed.

c. To definitively show that the induction of NF90/110 was only due to HG, control cells without HG should be included (or at least at the later time points).

A1-c. Thanks for your suggestion, our results showed that the induction of NF90/110 was only due to AGEs. We found that expression of NF90/110 in HAVSMCs began to increase on the third days after HG (27.5 mM) stimulation, and the peak of increased expression appeared on the fourth and fifth day of HG treatment (Fig. 1c, sup-Fig. 3a). Further AGEs assay displayed that the formations of AGEs in cell supernatant were also elevated obviously on the 4th and 5th day after HG stimulation (sup-Fig. 3b), and moreover the changes in NF90/110 began at 24 hours after AGE stimulation (sup-Fig. 3c). Therefore, we concluded that the induction of NF90/110 was only due to AGEs. Here, the HG 0 day and AGE 0 day were identified as the control cells without HG and AGE stimulation.

d. The authors only used male human specimens and male mice. The title and abstract do not indicate that the findings apply only to one sex. The method section also does not include whether sex and/or gender were considered in the study design.

A1-d. We appreciate the careful observation that the reviewer has made. We did omit the gender in the manuscript writing. Based on the considerable evidence suggesting that estrogen modulates cardiovascular physiology and function in both health and disease, and that it could potentially serve as a cardioprotective agent [1, 2, 3]. We considered the need to exclude the effect of gender at the beginning design of study and chose the male mice and specimens from male patients for this current study. As suggested by the reviewer, we indicated the gender of mice and patients in the abstract (page2, line27) and the methods section (page23, line467; page24, line495 and page25, line508).

Q2. Line 107. The reasoning is not totally accurate and should be revised. Two different mechanisms govern atherosclerotic vs CKD calcification, neo-intimal vs medial calcification, respectively.

A2. As suggested by the reviewer, we have revised the description in the manuscript (page5-6, line105-107).

Q3. Results section is too long (~16 pages), and it can be revised to be more concise.

A3. Thanks for your suggestion. For the sake of brevity, we modified the results section from 16 to 12 pages in text.

Q4. Since the study employed NF90/110 deficiency instead of overexpression, the word "... accelerates ..." should be revised in the title.

A4. We appreciate the careful observation that the reviewer has made. We have revised it (page1, line1-2).

Q5. Please revise the statement in line 519 since this study does not have evidence of increased RAGE expression.

A5. We have made the revision on page22, line450.

Q6. Although Figure 8 has improved, the following are needed for clarity.

1. Please mention in the legend how AGE enters into cells.

A6-1. We have made the revision on page56, line1142-1144.

2. Does AGE increase NF90/110 via RAGE signaling?

A6-2. This is an excellent point. In generally, whether AGEs increase NF90/110 via RAGE signaling was uncertain in current study. AGEs cause an activation of various

signaling pathways initiated by a series of receptors which including RAGE, AGER1, AGER2, AGER3, and CD36 et al [4, 5]. The most investigated human receptor is the type I cell surface RAGE belonging to the immunoglobulin (Ig) superfamily. Several other receptors for AGEs have also been identified and include the AGE-R1, -R2, and -R3 receptors and a group of scavenger receptors. RAGE initiates the intracellular signaling that disrupts cellular function through its recognition and binding of AGEs. Other receptors, like AGE-R1 (oligosaccharyl transferase-48), -R2 (80K-H phosphoprotein), and -R3 (galectin-3), and the class A macrophage scavenger receptor types I and II, also are able to recognize and bind AGE ligands, but they have not been shown to transduce cellular signals after engagement by AGEs. Instead, they may cause the clearance and possible detoxification of AGEs. Two class B scavenger receptors, CD36 and class B type I, also bind AGE ligands. CD36 is not involved in the clearance of AGEs from the circulation, but it plays an important role in the induction of oxidative stress in the cell. AGE ligands interfere with scavenger receptor class B type I uptake of HDL cholesterol. AGEs bind to and are recognized by the class E scavenger receptor, lectin-like oxidized LDL receptor-1 (LOX-1), and AGEs increase LOX-1 expression in diabetic rats [4, 5]. So, the receptor type that AGEs-mediated NF90/110 expression need be further explored.

3. It would be good to include panel titles in “a” and “b” (e.g. WT, NF90/110 KO, etc.).

A6-3. We have done that in Fig.8.

4. In panel a, please remove the arrow from “AGEs” to “More AGEs” and also remove the word “More AGEs.” Having the illustration of AGE (depicted as several circles) is self-explanatory.

A6-4. We have done that.

5. In the KO cells, please add more AGER-1 on the cell and also remove the word “Lower AGEs.”

A6-5. We have done that.

6. Please show a bigger difference AGE illustration (e.g 10 circles vs. 3) between the 2 panels.

A6-6. We have done that.

7. “Atherosclerotic” should be revised to “matrix calcification.”

A6-7. We have done that.

8. Panels a and b should depict differences in calcification.

A6-8. We have done that.

Minor comments

Q1. Methods for ALP and calcium content assay should be expanded. Please provide brief procedure, substrate name, OD wavelength, and how they were normalized, etc.

A1. We have done that on page31, line628-637.

Q2. Line 59 – Since calcium deposition is in the extracellular matrix, the statement should be revised “... stimulated calcium deposition in the extracellular matrix of ... VSMC ...”

A2. We have revised that on page3, line58-60.

Q3. Fig. 5C. The position of statistical bar in the graph should be properly aligned.

A3. We have done that.

Q4. Fig. 8. There seem to be typos in “NF100.”

A4. We apologize for the negligence. It has been revised.

Q5. Line 294 and 761. The word for the acronym RIP should be corrected to “RNA Immunoprecipitation.”

A5. We have corrected that in the full text.

Q6. Line 452... the word “deteriorate” does not fit the statement and should be revised.

A6. We have revised that on page19, line383.

Reviewer #2 (Remarks to the Author):

Q1. However, for Q14, removing the questionable supplemental Figure 4 without further adjustments is inappropriate.

A1. Thank you for the comment. the supplemental Figure 4 showed the co-expressions of NF90-FBXW7-AGER1 axis with osteoblastic markers OPN or pro-apoptotic markers Bax. It is an auxiliary result for Figure7 conclusion. In Figure7, western blot and immunohistochemical staining displayed the same conclusion in vivo and vitro. Considering your first comment, we thought that the supplemental Figure4 was too wordy. Moreover, the results section is too long, so, we removed the supplemental Figure 4 from the text.

Q2. Regarding to animal gender issue, there was no any information in the title and abstract. It was mentioned that male mice were used.

A2. As suggested by the reviewer, we remodified the abstract that indicated the sex (page2, line27), and the method section also indicated the sex of specimens (page23, line467; page24, line495 and page25, line508).

Q3. In addition, L354-L357 has a repeat sentence. Authors should check their MS more carefully.

A3. Thank you for the catch. We have deleted the repeat sentence and looked through the text carefully.

Citation

[1] Qinghai Meng, Yu Li , Tingting Ji, Ying Chao, Jun Li , Yu Fu, Suyun Wang, Qi Chen , Wen Chen, Fuhua Huang, Youran Wang, Qichun Zhang , Xiaoliang Wang. Estrogen prevent atherosclerosis by attenuating endothelial cell pyroptosis via activation of estrogen receptor α -mediated autophagy. J Adv Res. 2020 Aug 24;28:149-164

[2] Laila Aryan, David Younessi, Michael Zargari, Shadie Rahman, Reza Borna, Gregoire Ruffenach, Soban Umar, Mansoureh Eghbali. The Role of Estrogen Receptors in Cardiovascular Disease. Int J Mol Sci. 2020 Jun 17;21(12):4314.

[3] Monica De Paoli , Alexander Zakharia , Geoff H Werstuck. The Role of Estrogen in Insulin Resistance: A Review of Clinical and Preclinical Data. Am J Pathol. 2021 Sep;191(9):1490-1498.

[4]Alison Goldin, Joshua A Beckman, Ann Marie Schmidt, Mark A Creager. Advanced Glycation End Products Sparking the Development of Diabetic Vascular Injury. *Advanced Glycation End Products and Diabetes Mellitus: Mechanisms and Perspectives*. *Circulation*. 2006 Aug 8;114(6):597-605.

[5] Aleksandra Twarda-Clapa, Aleksandra Olczak, Aneta M Białkowska, Maria Koziółkiewicz. Advanced Glycation End-Products (AGEs): Formation, Chemistry, Classification, Receptors, and Diseases Related to AGEs. *Cells*. 2022 Apr 12;11(8):1312.

REVIEWER COMMENTS

Reviewer #1 (Remarks to the Author):

The authors have made significant improvements. Minor revision remains.

The title does not reflect the journal's requirement for sex since only male (mice and human specimens) was used. It should be revised as - NF90 deficiency in vascular smooth muscle cells ameliorates diabetic atherosclerotic calcification in male mice via FBXW7-AGER1-AGE axis

Line 467-468 – Please delete “specimens” and correct the spelling “Chines” to read as “... collected from Chinese males with or without”

Figure 8 legend is not clear and suggested as follows.

(A) In the NF90-sufficient diabetic condition, extracellular levels of AGE are increased due to reduced levels of AGER1, of which degradation is increased via the activated FBXW7/proteasome pathway, and VSMC-mediated matrix calcification is increased via the increased signaling of AGE-RAGE axis. (B) In the NF90-deficient diabetic condition, extracellular levels of AGE are decreased due to increased levels of AGER1, of which degradation is decreased via the inhibited FBXW7/proteasome pathway, and VSMC-mediated matrix calcification is decreased via the reduced signaling of AGE-RAGE axis.

Reviewer #2 (Remarks to the Author):

I have no further questions.

Itemized Responses to Reviewers Comments

Reviewer #1 (Remarks to the Author):

The authors have made significant improvements. Minor revision remains.

Q1. The title does not reflect the journal's requirement for sex since only male (mice and human specimens) was used. It should be revised as - NF90 deficiency in vascular smooth muscle cells ameliorates diabetic atherosclerotic calcification in male mice via FBXW7-AGER1-AGE axis

A1. Thank you very much, we have modified the title as your suggestion.

Q2. Line 467-468 - Please delete "specimens" and correct the spelling "Chines" to read as "... collected from Chinese males with or without ..."

A2. Thanks a lot, we have corrected the spelling in the text (page23, line 467-468).

Q3. Figure 8 legend is not clear and suggested as follows.

(A) In the NF90-sufficient diabetic condition, extracellular levels of AGE are increased due to reduced levels of AGER1, of which degradation is increased via the activated FBXW7/proteasome pathway, and VSMC-mediated matrix calcification is increased via the increased signaling of AGE-RAGE axis. (B) In the NF90-deficient diabetic condition, extracellular levels of AGE are decreased due to increased levels of AGER1, of which degradation is decreased via the inhibited FBXW7/proteasome pathway, and VSMC-mediated matrix calcification is decreased via the reduced signaling of AGE-RAGE axis.

A3. We have modified that as your suggestion.